# The underestimated role of stratosphere-to-troposphere transport on tropospheric ozone

Thomas Trickl<sup>1</sup>, Hannes Vogelmann<sup>1</sup>, Ludwig Ries<sup>2</sup>, Hans-Eckhart Scheel<sup>1,+</sup> and Michael Sprenger<sup>3</sup>

Correspondence to: Dr. Thomas Trickl, thomas.trickl@kit.edu, Tel. +49-8821-183-209, Fax +49-8821-73573

Abstract. The atmospheric composition is strongly influenced by changing atmospheric dynamics, in potential relation to climate change. A prominent example is the doubling of the stratospheric ozone component at the

- summit station Zugspitze (2962 m a.s.l., Garmisch-Partenkirchen, Germany) between the mid-seventies and 2005, roughly from 11 ppb to 23 ppb (43 %). Systematic efforts for identifying and quantifying this influence have been made since the late 1990s. Meanwhile, routine lidar measurements of ozone and water vapour carried out since 2007, combined with in-situ and radiosonde data and trajectory calculations, have revealed the presence of stratospheric intrusion layers on 84 % of the yearly measurement days. The seasonal cycle for deep
- intrusions with a pronounced summer minimum seen at Alpine summit stations disappears if one looks at the entire free troposphere. The seasonal cycle previously obtained for the Zugspitze summit is rather well reproduced by the lidar data. The mid- and upper-tropospheric intrusion layers seem to be dominated by very long downward transport up to a full tour around the northern hemisphere in an altitude range starting at about 4.5 km a.s.l. Unless there is a strong perturbation, these layers remain considerably dry, typically with RH  $\leq$  5 %
- at the centre of the intrusion. It is interesting to note that, in recent years, most pronounced ozone maxima have been related to a stratospheric origin rather than to long-range transport from remote boundary layers. This fact could be caused by improving air quality in the most relevant source regions or changing transport patterns.

Key words: Ozone, water vapour, aerosol, stratosphere-to-troposphere transport, transport modelling, lidar, LAGRANTO

# 30 1 Introduction

Quantifying stratosphere-to-troposphere transport (STT) has been attempted as long as over more than half a century. After early studies to identify the general mechanisms (e.g., Danielsen, 1968) more recent work has also estimated of the STT budget by extrapolations of observational data (e.g., Danielsen and Mohnen, 1977; Viezee et al., 1983; Beekmann et al., 1997), and by modelling efforts (e.g., Roelofs and Lelieveld, 1997; Kentarchos and

Roelofs, 2003; Stevenson et al., 2006; Wild, 2007; Young et al., 2013). There is, still, a considerable uncertainty of the results. The global distribution is rather inhomogeneous with maxima in regions around the jet-stream belts (James et al., 2003; Sprenger et al., 2003; Škerlak et al., 2014)). Sprenger et al. (2003) found that the role of

 <sup>&</sup>lt;sup>1</sup>Karlsruher Institut für Technologie, Institut für Meteorologie und Klimaforschung, IMK-IFU, Kreuzeckbahnstr.
 19, 82467 Garmisch-Partenkirchen, Germany
 <sup>2</sup>Umweltbundesamt II 4.5, Plattform Zugspitze, GAW-Globalobservatorium Zugspitze-Hohenpeißenberg, Schneefernerhaus, 82475 Zugspitze, Germany
 <sup>3</sup>Eidgenössische Technische Hochschule (ETH) Zürich, Institut für Atmosphäre und Klima, Universitätstraße 16, 8092 Zürich, Switzerland
 <sup>4</sup>Deceased on June 23, 2013

the subtropical jet stream (STJ) for STT had been strongly underestimated. The STJ persists during most of the year, but it is not only the persistence that matters: Very strong ozone signatures in the troposphere exceeding 200 ppb have been detected over northern India (Ojha et al., 2014; 2017). High ozone values in the middle and upper troposphere from regions next to the STJ have even been observed above Garmisch-Partenkirchen (Germany) after transport almost all the way around the northern hemisphere (Trickl et al., 2011).

- (Germany) after transport almost all the way around the northern hemisphere (Trickl et al., 2011). An approach to quantify the fraction of stratospheric ozone based on observations alone was made by Scheel (pp. 66-71 in (ATMOFAST, 2005)) for the lower-tropospheric site Zugspitze (2962 m a.s.l.) in the German Alps. Here, ozone rose from 1978 to about 2003, in contrast to the neighbouring Wank site (1780 m a.s.l.) where constant ozone was observed since the 1980s. Elbern et al. (1997) found that the higher Zugspitze station
- registered more than twice the stratospheric air intrusions than the lower Wank site which is in agreement with the trend difference. Scheel also detected that the only significantly growing ozone component in the 1990s was associated with very dry air that had obviously descended from high altitudes (Scheel, 2003). By correlating ozone with <sup>7</sup>Be and low relative humidity (RH) STT events can be clearly identified ("direct intrusions"). A positive trend of the direct stratospheric component in Zugspitze ozone was found even back to the beginning of
- the high-quality measurements at the Zugspitze summit in 1978, the contribution, however, being below 10 % (Fig. 1). The indirect component that cannot be identified by data filtering was estimated from <sup>7</sup>Be considering that 2/3 of <sup>7</sup>Be had been reported to be of stratospheric origin (Lal and Peters, 1967). <sup>7</sup>Be was converted to ozone by calibration with the direct component. Both stratospheric components in Fig. 1 grew from 1978 to 2003, the clearest evidence of an increasing STT contribution coming from <sup>7</sup>Be that started to rise in the mid-seventies. In
- 2003, the stratospheric ozone component reached an astonishing value of 23.4 ppb (i.e., 43.4 % of the 2003 annual average). This result seems to confirm the average 40 % obtained by Roelofs and Lelieveld (1997) for the entire troposphere. <sup>7</sup>Be has been measured at the Zugspitze summit since late 1969 (Reiter et al., 1971). This series reveals that the positive trend started in the mid-seventies. A comparable positive ozone trend is also reported for the Swiss Jungfraujoch station (3580 m a.s.l.) where the measurements started in 1992 (Ordoñez et
- al., 2007). For the lower-lying Italian station Monte Cimone (2165 m a.s.l.) the absence of a significant correlation between ozone and the intrusion frequency was concluded, but for the years 1996 to 2011, that are rather late compared with the main rise at the Alpine sites (Cristofanelli et al., 2015). Scheel's approach is still associated with an uncertainty of unknown magnitude, in part because of the limited

decay time of this isotope (53.42 (±0.01) days; Huh and Liu, 2000). However, the result cannot be too wrong

- since the 1978 stratospheric fraction of 11.3 ppb (31.2 %) is in the range of expectations from the Montsouris value derived from measurement in the late 19<sup>th</sup> century (Volz and Kley, 1988). In addition, after subtracting the stratospheric contribution the positive trend disappears. From the decline of the ozone precursors in the 1990s (e.g., Jonson et al., 2006; Vautard et al., 2006) one would have even expected a negative development. Thus, the result is perhaps even somewhat conservative.
- The most reasonable explanation in our view has been a reaction of the atmospheric dynamics to the climate change. Indeed, several authors have given hints on this (e.g., Collins et al., 2003; Lin et al., 2015; Neu et al., 2014).

Trickl et al. (2010) analysed the data-filtering criteria for STT events reaching the Zugspitze summit. By comparison with transport model predictions and backward trajectories they found that the criteria used (also in

Fig. 1) are highly reliably for identifying stratospheric air masses. However, the <sup>7</sup>Be thresholds previously used

were found to be far too high. Thus, it is planned to revise the calculations and to look for a potential positive trend of the annual intrusion count. In view of a full quantification one open question had been the influence of tropospheric mixing of the descending air layers. Lidar measurements revealed that mixing is rather small and in many cases almost negligible (Trickl et al., 2014; 2015; 2016).

- 5 The seasonal cycle of ozone at Alpine summit stations exhibits a pronounced summer minimum (Elbern et al, 1997; Stohl et al., 2000; Trickl et al., 2010). By contrast, Beekmann et al. (1997) concluded for the entire free troposphere above three European ozone-sonde stations a seasonal cycle with a slight summer maximum, based on data filtering of ozone profiles between 1969 and 1994. A transition to this behaviour is indicated for growing altitude of the Alpine stations: The summer minimum is least pronounced at the highest of the stations previously compared, Jungfraujoch (see Fig. 1 of Trickl et al., 2010).
  - In this paper, we extend our STT studies to the full free troposphere. The analysis is based on routine lidar measurements of ozone, water vapour and aerosol since 2007, as well as radiosonde relative humidity and transport modelling. These results reveal a surprisingly strong stratospheric component in tropospheric ozone over Central Europe.

# 15 2 Methods

# **2.1 Measurements**

#### 2.1.1 IFU ozone DIAL

The tropospheric ozone lidar is operated in Garmisch-Partenkirchen, Germany at IMK-IFU (formerly IFU; 47° 28′ 37" N, 11° 3′ 52" E, 740 m a.s.l.). The laser source is a Raman-shifted KrF laser, and two separate receiving

- telescopes are used to divide the dynamic range of the backscatter signal of roughly eight decades. This lidar was completed as a two-wavelength differential-absorption lidar (DIAL) in 1990 (Kempfer et al., 1994) and a first annual sounding series was achieved in 1991 (Carnuth et al., 2002). It was upgraded to a three-wavelength DIAL in 1994 and 1995 (Eisele and Trickl, 1997), leading to a unique vertical range between roughly 0.25 km above the ground and 3 to 5 km above the tropopause, the measurement time interval being just 41 s. By comparing the
- ozone profiles retrieved from different wavelength combinations (e.g., 277 nm 313 nm or 292 nm 313 nm an internal quality check is possible. The choice of an "on" wavelength below 280 nm is particularly beneficial for achieving a high accuracy and a high vertical resolution (up to about 5 km above the ground). Density has been converted to mixing ratio by using pressure and temperature data from nearby radiosonde stations (Sect. 2.1.4).
- The DIAL features low uncertainties of about ±3 ppb in the lower free troposphere, approximately doubling 30 (under optimum conditions) in the upper troposphere. Comparisons with the nearby Zugspitze in-situ measurements (at 2962 m a.s.l., see below) show no relevant mutual bias, the standard deviation of the differences being less than 2 ppb. The uncertainty further diminished after another system upgrading in 2012, after introducing a new ground-free input stage to our transient digitizers (Licel) that reduced the noise level by roughly a factor of three. For the range covered by the near-field receiver (below 1.2 km above the lidar) the
- uncertainty is of the order of ±5 ppb. The upper-tropospheric performance may be degraded in the presence of high lower-tropospheric ozone concentrations absorbing a lot of the ultraviolet laser emission and by enhanced sky light in summer, in particular in the presence of clouds. Thus, longer data acquisition times, requiring some technical modifications, are planned for the future. The vertical resolution is dynamically varied between 50 m

and a few hundred metres, depending on the signal-to-noise ratio decreasing with altitude. Within stratospheric intrusion layers the vertical resolution is reduced as far as possible in order to avoid a reduction of the peak concentrations by smoothing. The lidar has been used in numerous atmospheric transport studies (e.g., Eisele et al., 1999; Stohl and Trickl, 1999; Trickl et al., 2003; and other publications cited in this paper).

- Aerosol backscatter profiles with very good signal-to-noise ratio up to the lower stratosphere are obtained from the 313-nm "off" channel of the lidar. The methods, implying an ozone correction, have been described by Eisele and Trickl (2005). Examples demonstrating the data quality achieved in recent years (maximum noise level of the backscatter coefficients: ±1×10<sup>-7</sup> m<sup>-1</sup> sr<sup>-1</sup>, reached in the tropopause region) can be seen in (Trickl et al., 2015). We derive vertical profiles of the aerosol backscatter coefficients based on on a constant
- backscatter-to-extinction ratio of 0.020 sr<sup>-1</sup>, which is the average value derived within the European Aerosol Research Lidar Network (EARLINET, 2003). Within clouds larger values are taken, if possible optimized for minimum discrepancy of the backscatter profiles below and above the cloud.

# 2.1.2 IFU water-vapour DIAL at the Schneefernerhaus high-altitude station

The Zugspitze water-vapour DIAL is operated at the Schneefernerhaus high-altitude research station (UFS, 47°
25' 00" N, 10° 58' 46" E) at 2675 m a.s.l., about 8.5 km to the south-west of IMK-IFU (Garmisch-Partenkirchen, Germany), and 0.5 km to the south-west of the Zugspitze summit. The full details of this lidar system were described by Vogelmann and Trickl (2008). It is based on a powerful tunable narrow-band Ti:sapphire laser system with up to 250 mJ energy per pulse operated at about 817 nm and a 0.65-m-diameter Newtonian receiver. Due to these specifications a vertical range up to about 12 km can be reached, almost unaffected by daylight.

- However, mostly the laser has been operated at half the maximum pulse energy or less to extend the life time of the high-voltage components such as flashlamps. A separation of near-field and far-field signals is achieved by a combination of a beam splitter and a blade in the far-field channel. The operating range starts below the altitude of the summit station (2962 m a.s.l.). The electronics are almost identical to those of the ozone DIAL. However, at the operating wavelength of 817 nm avalanche photodiodes have been used that introduce higher noise than
- the photomultiplier tubes preferred for shorter wavelengths. Thus, the system has not yet reached its expected optimum performance in the upper troposphere. The vertical resolution chosen in the data evaluation is dynamically varied between 50 m in altitude regions with good signal-to-noise ratio and roughly 350 m in the upper troposphere. Free-tropospheric measurements during dry conditions clearly benefit from the elevated site outside or just below the edge of the moist Alpine boundary
- layer (e.g., Carnuth and Trickl, 2000; Nyeki et al., 2000; Carnuth et al., 2002). After a few years of testing, validating and optimizing the system routine measurements were started in January 2007 with typically two measurement days per week, provided that the weather conditions are favourable. Operation has been interrupted since Winter 2015 due to fatal laser damage. A new Ti:sapphire laser system is under development.

The lidar has been validated in several comparisons with local and remote radiosonde ascents (Vogelmann and

35 Trickl, 2008), an airborne DIAL (Trickl et al., 2016) and the Zugspitze Fourier-transform spectrometer (Vogelmann et al. 2011). A noise level of 5 % and a bias of 1 % at most was verified to more than 6 km. Furthermore, a very high importance of volume matching in comparisons of water-vapour profiling instruments was found (see also (Vogelmann et al, 2011; 2015), on the scale of a quarter of an hour and a few kilometres.

In some cases in which a direct comparison of the exact matching of the humidity and aerosol layers was necessary (e.g., Trickl et al., 2016) aerosol backscatter coefficients were retrieved from the "off" wavelength channel. The calculations were done with a program developed for the IFU aerosol lidar systems (e.g., Trickl et al., 2013; Wandinger et al., 2016).

## 5 2.1.3 In-situ measurements at the Zugspitze summit and at the Schneefernerhaus station (UFS)

In addition, in-situ data from the monitoring station at the Zugspitze summit (air inlet: 2962 m a.s.l.) have been inspected, namely ozone and relative humidity. Ozone was measured between 1978 and 2012 (e.g., Reiter et al., 1987; Scheel et al., 1997; Oltmans et al., 2006; 2012; Logan et al., 2012; Parrish et al., 2012). At present the data have been evaluated until 2010. The relative uncertainty of the Zugspitze ozone is 1 %. Ultraviolet absorption

instruments have been employed (Thermo Electron Corporation, U.S.A., TE49 analysers). Relative humidity (RH) was registered with a dew-point mirror (Thygan VTP6, Meteolabor, Switzerland) with a quoted uncertainty below 5 % RH. However, the instrument has a wet bias of almost 10 % under very dry conditions (Trickl et al., 2014). The Zugspitze measurements were discontinued in January 2013.

After the end of these measurements we have used the corresponding data of the Global Atmosphere Watch

- (GAW) observatory at the Schneefernerhaus research station (UFS, see H<sub>2</sub>O lidar), operated by the German Umweltbundesamt (UBA, i.e., Federal Environmental Agency; 47° 25′ 0″ N, 11° 58′ 46″ E; air inlet at 2670 m a.s.l.). Ozone is continuously measured by ultraviolet (UV) absorption at 254 nm (Thermo Electron Corporation, model Ts49i). Relative humidity is measured by the German Weather Service with an EE33 humidity sensor (E+E Elektronik). The calibration of the UBA instrumentation is routinely verified as a part of the GAW quality
- assurance efforts. The instruments are controlled daily and serviced on all regular work days. For the comparisons shown in the figures of this paper we use time averages up to one hour because of the time delay of the air mass between UFS and IFU. In the presence of nearby vertical steps in the ozone distribution orographic lifting must be taken into consideration that can lead to vertical displacements.

#### 2.1.4. Sonde data

more remote sites have been inspected.

35

- 25 Radiosonde data are routine used for calculating the atmospheric density, which is necessary for quantitative aerosol retrievals and the conversion of the ozone or water-vapour number density to mixing ratio. Most importantly, on each measurement day of the ozone DIAL the presence of dry and moist layer was examined in view of an identification advection from a remote stratosphere or (marine) boundary layer. The sonde data have been imported from the University of Wyoming data base (http://weather.uwyo.edu/upperair/sounding.html).
- 30 Preferentially, the Oberschleißheim ("Munich") sonde RH has been examined, this station (number: 10868) being located 100 km roughly to the north. If data were not available for a given time or if no indication of an intrusion was found RH profiles from other surrounding stations were used such as Stuttgart (10739, about 200 km to the north-west), Payerne (06610, about 310 km to the west), or Innsbruck (11120, 32 km to the south, one measurement per day only). In critical cases the station choice was also based on the trajectory results, and even

The sonde type used by the German Weather Service (DWD, Deutscher Wetterdienst) during the period presented here was RS 92 (Vaisala; e.g., Miloshevich et al., 2006; Steinbrecht et al., 2008). The sonde data

feature an artificial cut-off at 1 % for conditions when the UFS DIAL revealed even even much drier conditions (Trickl et al., 2014).

#### 2.2 Models

# 2.2.1 LAGRANTO

- Four-day forward trajectories have been calculated once a day for start times t<sub>0</sub> = 1:00 CET (Central European Time, = UTC + 1 h), t<sub>0</sub> + 12 h, t<sub>0</sub> + 14 h and t<sub>0</sub> + 36 h based on the Lagrangian Analysis Tool (LAGRANTO; Wernli and Davies, 1997a; Sprenger and Wernli, 2015) since summer 2000. On each day, trajectories are calculated using operational forecast data from the European Centre for Medium-Range Weather Forecasts (ECMWF) interpolated to with 1°×1° horizontal resolution. For each start time four-day forward trajectories are
- calculated, starting in the entire region covering the Atlantic Ocean and Western Europe (20° east to 80° west and 40° to 80° north) between 250 and 600 mbar. From this large set of trajectories those initially residing in the stratosphere (potential vorticity larger than 2.0 pvu) and descending during the following four days by more than 300 mbar into the troposphere were selected as "stratospheric intrusion trajectories". The same selection criterion was used in a previous case study (Wernli, 1997b) to study an intrusion associated with a major North Atlantic
- cyclone.

Since June 2001 so-called "intrusion hit tables" have been additionally distributed giving a crude estimate of the time-height development of stratospheric air above the four STACCATO (Stohl et al., 2003) partner stations Jungfraujoch, Zugspitze, Monte Cimone and Thessaloniki over several days. Both the STT trajectories and the hit tables are daily distributed to all interested partners and institutions. Intrusion warnings based on these images have been issued by IFU if several of the stations could be affected (Zanis et al., 2003b).

images have been issued by IFU if several of the stations could be affected (Zanis et al., 2003b). For special case studies LAGRANTO has been operated with re-analyses meteorological data, for periods up to five days (e.g., Trickl, 2014; 2016). The three-dimensional wind fields for the calculation of the trajectories were taken from ERA-Interim data set (Dee et al., 2011) from ECMWF, which was interpolated to a with 1×1 horizontal resolution and provides winds at 6-h intervals.

# 25 2.2.2 HYSPLIT

For analysing intrusion events with travel times exceeding the four days set in the LAGRANTO forecast runs or with source regions outside the domain of the forecasts we use HYSPLIT (Hybrid Single-Particle Lagrangian Integrated Trajectory, Draxler and Hess, 1998; http://ready.arl.noaa.gov/HYSPLIT.php) backward trajectories. The three-dimensional ("model vertical velocity") backward trajectory calculations were preferentially based on

- re-analysis meteorological data. Although the re-analysis data are coarser than other meteorological data available they have led to a superior model performance in the free troposphere in many of our studies (Trickl et al., 2010; 2013; 2014; Fromm et al., 2010) and the analysis of our routine measurements: Despite the known limitations of backward trajectories (e.g., Stohl and Seibert, 1998) most specific free-tropospheric layers in years of observations could be related to reasonable sources with this operation mode of HYSPLIT, the best
- investigated transport type being STT. Since 2014 near-real-time data evaluation and aerosol archiving in the EARLINET (European Aerosol Research Lidar Network, https://www.earlinet.org) data base have been achieved. Thus, GDAS-based trajectories (GDAS: Global Data Assimilation System) have been taken since the

re-analysis-based model version are available only with considerable delay. In a number of cases the GDASbased trajectories did not verify STT. However, later runs in the re-analysis mode for the present study clearly verified a stratospheric origin of the air mass of interest (see also Trickl et al., 2015).

The calculations were extended to the full 315-h period supported by the model. In very rare cases the trajectories did not end in an altitude range corresponding to expectations for the lower stratosphere in the outflow region of an intrusion (e.g., roughly 7.5 km or more in boreal regions) within 315 h. In some these cases extension trajectories were calculated to verify the stratospheric source (Trickl et al., 2015). Otherwise, these cases were rejected.

Slight vertical displacements of intrusion layers at the northern rim of the Alps exist in the model runs as reported previously (Trickl et al., 2010; 2015). These offsets, that vary from case to case, are explained by the insufficiently resolved orography that leads to an altitude of IMK-IFU (730 m a.s.l.) roughly half-way between the valley (Garmisch-Partenkirchen) and the Zugspitze summit. It is important to activate the check box "terrain" on the HYSPLIT input page together with the altitude option "AMSL" (above mean sea level). In this case, the absolute height is used on the vertical axis as well as the contour of mountains are displayed, and better

agreement with the altitude of an arriving atmospheric layer is achieved. The trajectories are lifted above mountain ranges, which is particularly spectacular above Greenland with a surface altitude of about 3 km maintained over hundreds of kilometres.

#### **3** Results

#### 3.1 Description of the data analysis and interpretation

- Starting in 2007, routine measurements have been started with both DIAL systems. This has yielded vertical profiles of ozone, water vapour and aerosol backscatter coefficients, derived from the 313-nm channel of the ozone DIAL. The number of measurements is particularly high in the case of the ozone lidar, resulting in a total of 2275 evaluated data files between 2007 and 2016 (Table 1). The present study is, therefore, based on the ozone soundings during this period and all other data are used for identifying the source for conspicuous ozone
- structures such as stratospheric air intrusions. Measurements have been made on a large number of fair-weather days or during short periods of clearing. However, really strong efforts to make at least one measurement were limited to the EARLINET (European Aerosol Research Lidar Network) "climatology days" Monday and Thursday (EARLINET, 2003). Ancillary information from sondes and trajectories has been gathered for each measurement day.
- There are several gaps in the data. These gaps are explained by extended periods of laser or computer damage, sometimes involving the search for new technical solutions for the system. The latest one occurred between August 2016 and September 2017. The temporal co-incidence of ozone and H<sub>2</sub>O measurements had gradually grown before the fatal laser damage of the Ti:sapphire laser in early 2015 (a new laser is under development).
- Layers with elevated ozone were analysed either for the existence of a positive forecast with LAGRANTO and/or with HYSPLIT trajectories. For HYSPLIT, potential vorticity has not been available for identifying the tropopause and, thus, descent from roughly 7.5 km ore more (preferentially at higher latitudes) was looked for. This altitude had been found to be sufficient from previous analyses and the LAGRANTO calculations that select stratospheric trajectories based on potential vorticity. With decreasing latitudes and in summer higher start

altitudes preferred. At the same time low lidar or sonde relative humidity (RH) with minimum values clearly below 10 % had to co-exist at least at adjacent altitudes. This was fulfilled for most cases, 8-10 % RH being the exception. In confirmation of our results from the water-vapour DIAL (Trickl et al., 2014; 2015; 2016) we found in the sonde data typical minimum RH values of 1-2 % for source regions over the North Atlantic (intrusion

- 5 Types 1-5 as defined by Trickl et al. (2010)), 1 % being the lowest value found in the sonde listings (Sect. 2.1.4). For long-range descent from a remote stratospheric source (e.g., western Canada, Alaska, Siberia: Type 6) or slow descent from the North Atlantic minimum RH mostly ranged between 3-6 %. It is interesting to note that, reversely, finding RH values in this range in sonde data with great reliability led to very long transport times. Quite surprisingly, the longest descent analysed (15-17 days) led to negligibly low H<sub>2</sub>O in the DIAL measurements at UFS (July 16, 2013: Trickl, 2015).
  - Intrusions reaching altitudes around 3 km were verified by looking at the Zugspitze ozone and RH data until 2010 and the UFS data afterwards. As pointed out in Sect. 2.1.3 the Zugspitze RH rarely dropped clearly below 10 %.

Range of ozone values in intrusion layers was between 5-10 % above the background (last figure of Trickl et al.,

- 2014) and 235 ppb (see below). The smallest enhancements could be identified due to the strongly improved data quality of the ozone DIAL. Just a few cases had to be rejected either due to insufficient dryness or not not fully conclusive trajectory results. Sometimes these weak structures were looked for only after finding strong evidence of a very dry layer in the daily inspected humidity data and after a verification with trajectories. In winter, in the absence of strong photochemical ozone production, the existence of a pronounced ozone peak in
- the vertical profile is, of course, highly indicative of the presence of stratospheric air.

#### 3.2 Typical findings

As previously discussed (Trickl et al., 2010) stratospheric air intrusions passing over Garmisch-Partenkirchen arrive from almost all directions. Easterly directions mostly result from detours of the dry layers via Eastern Europe or curl formation over Central Europe potentially in cut-off lows. A few HYSPLIT results some rather complex pathways.

complex pathways.

It is interesting that intrusion layers can be observed under so many different conditions. We routinely observe pre-fontal and post-frontal intrusion layers, as well as intrusions slowly descending from the far west. The first prefrontal ozone peak was detected above Garmisch-Partenkirchen just ahead of the 1997 VOTALP Föhn case (Seibert et al., 2000). These cases are frequently associated with a descent of the stratospheric air masses from

- the Arctic to North Africa followed by some return to Central Europe, accompanied by Saharan dust. These layers normally rise as a function of time as they are on a transition into a warm conveyor belt (e.g., Cooper et al., 2004). Postfrontal intrusions have been more frequently studied. They mostly reach low altitudes above Garmisch-Partenkirchen and occur very reliably, of course in the "classical" case of beginning anti-cyclonic conditions (e.g., Stohl and Trickl, 1999; Trickl et al., 2003), but even between two fronts sometimes separated by
- not more than one day. In these cases the inclined descending layer can be sandwiched between the low-lying clouds of the preceding front and the high-lying clouds of the incoming new front.

20

25

A few specific remarks:

(1) Very intense intrusions have been rare

Although intrusions with 100-150 ppb of ozone in the middle and upper troposphere are not that rare much higher values are really exceptional. Just three cases with peak ozone mixing ratios reaching or exceeding

5 200 ppb have been found during the period described here. The most intense intrusion case was observed on 1 October 2015 (Fig. 2). The peak ozone mixing ratio on that day was 235 ppb. The high ozone values did not last long. The peak ozone values rapidly dropped to less than 100 ppb. For comparison we give average values of the in-situ GAW measurements at UFS (2670 m). The UFS data exhibit a slight negative bias of 2 to 5 ppb which, given the normally better agreement, is ascribed to orographic lifting of the air masses that arrived from the east (see below), along the former glacier basin.

The presence of an intrusion on 1 October 2015 had been predicted by the LAGRANTO operational forecasts. For this paper the trajectory calculations were repeated with re-analysis data (ERA Interim) at intervals of just six hours and extended to five days. In Fig. 3, we give one example for a start time of 12 UTC (13 CET) on 25 September. This start time is slightly too early with respect to the lidar observations.

15 However, the trajectory plots for the later times become more and more complex with streamers for quite different arrival times over Central Europe confusing the picture. The structures in Fig. 3 persist in addition to the new trajectory bundles.

Figure 3 and later plots show high-lying and low-lying trajectories over South Germany and qualitatively confirm the observations. Some more clearness comes from the HYSPLIT backward trajectories that allow one to separate in time and altitude above the lidar. The HYSPLIT calculations (not shown) nicely confirm the main pathway shown in Fig. 3 with a northward departure over Greenland followed by a decent via Eastern Europe.

- (2) 235 ppb were also observed on 26 February 2015 (at 7 km), and a particularly spectacular case with 200 ppb was reported for March 6, 2008 (Trickl et al., 2014). *Extremely thin layers survive the long-range transport with almost negligible mixing with tropospheric air* 
  - The width of intrusion layers can vary considerably from case to case. Layer widths exceeding 1 to 2 km, in particular that in Fig. 2, are rare. However, the simultaneous presence of several dry layers is quite frequent as shown in our earlier work (e.g., Zanis et al., 2003b; Trickl et al., 2010). Also very thin layers with widths of down to 0.2 km have been observed. Both IFU DIAL systems are capable of resolving these structures,
- 30 and there is mostly very little mixing with tropospheric air (Trickl et al., 2014; 2015; 2016). A particularly spectacular case (26-27 December 2008) was discussed by Trickl et al. (2014) and verified by high-resolution transport modelling. Here, we show as an example two parallel thin high-ozone layers descending to Alpine summit levels (Fig. 4).
  - (3) Slow long-distance descent (Type 6) dominates the observations above about 4.5 km
- The slow descent of stratospheric layers from remote source regions such as Western Canada, Alaska or even Siberia down to Alpine summit levels was identified by Trickl et al. (2010; 2015). These Type 6 intrusions (definition: Trickl et al., 2010) are shown here to become even more important if one looks at the entire free troposphere. A source can also be the subtropical jet stream on its way across Asia, reaching midand high latitudes over the Pacific Ocean (Trickl et al., 2011).

5

- (4) Intrusion layers arriving via North Africa ahead of a long front are mostly not mixed with Saharan dust because they are typically located above the dust layer (exceptions exist). In Fig. 5 we show time series of the ozone aerosol profiles on 31 January and 1 February 2014, and Fig. 6 displays three HYSPLIT trajectories selected for the three relevant ozone and aerosol layers at midnight between the two days. The dust was lifted to roughly 5 km which is typical of these Föhn events at the northern rim of the Alps (e.g., Jäger et al., 1988; Papayannis et al., 2008). Above the dust layers a layer with elevated ozone passed over our site. The minimum Munich RH on 1 February at 1 CET was 2 % indicating a moderate travel time (the UFS DIAL was not operated). The intrusion trajectory in Fig. 6 shows rather rapid travel from about 10 km
- above Cape Farvel (Greenland) to North Africa. Obviously, this high speed makes it possible to pass eastward over the southern part of the front and enter the air stream rising to the Alps. The HYSPLIT trajectories for other altitudes around 7 km start to differ strongly vertically and horizontally upstream Eastern Canada. The LAGRANTO STT forecast confirms at least transport from Labrador to North Africa. Another dust case this time combined with long-range descent from the Northern United States (U.S.) (and presumably beyond) and a single-loop curl at low latitudes (see (7)) is shown in Fig. 7 (18 June 2013), with
- 15 a corresponding trajectory analysis in Fig. 8. Here, the intrusion air mass crossed the cold front over Western Spain or Portugal. At 13 CET on 18 June this front extended north-south from Bristol (U.K.) to North Africa. Obviously, the upper end of the clouds was rather low in this area, similar to the May-1996 case in (Trickl et al., 2003). The RH determined from a comparison of the results of the water-vapour DIAL (right panel in Fig. 7) and the Munich radiosonde was 8 to 12 %. These rather high values are ascribed to the very long descent over at least thirteen days.

Intrusions co-existing with Saharan dust have been observed on a total of 67 days. The number of dust days in our record is limited because frequently dust arrives below clouds and not always lidar measurements have been made.

- (5) Elevated summer-time ozone above about 4.5 km was a feature observed many times. Two examples from May and August 2015 are shown in Fig. 9. In both cases (and most others) dry layers exist within the highozone range and the corresponding HYSPLIT trajectories stay at high altitudes. For the altitudes analysed with trajectories we did not find any contact with potentially polluted planetary boundary layer (PBL) within the maximum of 315 h provided by the model. However, the calculation for a cirrus layer around 7.5 km on 10 August ended around 5 km over the Pacific, giving some hint on the origin of the moisture.
- 30 The quality of the lidar measurements in the upper troposphere was reduced due to the elevated absorption of the radiation by ozone. The vertical range was slightly larger during the darker hours.
  - (6) Volcanic and fire aerosol are transported downward from the lowermost stratosphere

During the periods of major volcanic activity impacting the lower stratosphere frequently aerosol was detected in intrusion layers (see also Browell et al., 1987). More specifically, particles in intrusion layers

35 were registered after the eruptions of Okmok and Kasatochi (July and August 2008, respectively; see (Trickl et al., 2016)), Redoubt (March 2009), Sarychev (June 2009) and Nabro (June 2011) (more details: Trickl et al., 2013). Typical 313-nm aerosol backscatter coefficients were 5×10<sup>-7</sup> m<sup>-1</sup> sr<sup>-1</sup> and less. The highest value was 2.35×10<sup>-6</sup> m<sup>-1</sup> sr<sup>-1</sup>, observed on 7 September 2009, after the viol