# Peer review of "The underestimated role of stratosphere-to-troposphere transport on tropospheric ozone"

_Atmospheric Chemistry and Physics, 2017_

## Referee Comment (RC1) · Anonymous Referee #1 · 6 Mar 2018

The manuscript "The underestimated role of stratosphere-to-troposphere transport on tropospheric ozone" by Trickl et al. describes an analysis of the effect of stratospheric intrusions over Garmisch-Partenkirchen on ozone values over the full free troposphere. Long-term ozone and water vapor lidar measurements from Garmisch-Partenkirchen, in-situ and radiosonde data from nearby stations, and trajectory analyses combined provide a very good data basis for these kinds of analyses. Different intrusion events are categorized, and possible reasons for changes in event frequencies are mentioned. The structure of the manuscript is somewhat chaotic so that it is difficult to find the storyline in the text. Sometimes the sentence structures are not entirely correct which complicates the understanding of the meaning of phrases and sentences. I believe the applied analyses methods are sound, however, it is hard to tell since there is an

abundance of information about historical analyses already published, recent analyses, and planned analyses. I think this manuscript needs major restructuring, rewording and general clarifications before the full extent of the presented analyses becomes clear. But I do believe that the study can show some interesting results if they are presented in a precise and detailed manner. I can only recommend the manuscript for publication if major revisions have been done, following the general and detailed suggestions listed below.

General suggestions/comments:

- The manuscript has both too much and not enough information in parts. Very often, much background information about measurements, specific intrusion cases etc. are given that might not be necessary for the storyline (some of these will be pointed out in the specific comment section). And then there are crucial information parts missing to be able to understand which exact intrusion case is actually referred to, or which reference is referred to (some of these will be pointed out in the specific comment section). I strongly recommend checking the whole manuscript for these things to enhance the clarity of the analysis and the storyline. At the moment the storyline (what was exactly done, why was it done, how was it done) is not obvious, which makes reading the manuscript and understanding its message very difficult.

- The manuscript would be easier to read if the authors would refer to specific references, measurement systems or intrusion cases in the same way. For example on page 14, the reference 'Beekmann et al. (1997)' is discussed. However, within this paragraph, this study is also referred to as 'the 1997 analysis' or 'the 1997 results'. Repeating the reference as 'Beekmann et al. (1997)' is not an unnecessary repetition of words, but actually very helpful in understanding what is discussed in the paragraph. Please go through the manuscript and standardize the description of the same reference/results/methods.

- Very often just one or two words are missing to clarify the meaning of a sentence or

a phrase. For example, on page 14, line 7, 'seasonal cycle peaks' is not unambiguous. 'Season cycle peaks' of what exactly? Some more of the ambiguous descriptions will be provided in the special comments section, but I encourage the authors strongly to check the manuscript independently for these ambiguous descriptions.

- Abstract: The first two sentences are more relevant to an introduction than an abstract. This might be the motivation for the study, but does nothing to explain exactly what has been done in the analysis. Please rewrite the abstract to include all the important parts of information necessary to understand the performed analysis (for example, the exact time period on which the analysis is based), and also state very clearly the main findings of the study.

- The main message of the manuscript is buried in many details and case studies. I think it is necessary to restructure the manuscript that the results of the main analyses become clearer. You present systematic analysis of stratospheric intrusions over the whole troposphere; you discover a change in the annual cycle of intrusion frequencies when you do this; there are many more intrusion layers than previously reported (did I understand this correctly?); some of the changes might be due to a change in climate. If these are the main points of your study, make sure that every reader will understand this and the connections between these points.

- The use of 7Be as tracer for descending air is explained in detail at the beginning of the manuscript. However, in the discussion section it is mentioned that analyses including 7Be are only planned for the future and are not done yet. Why the lengthy explanation about this method then earlier in the document?

- Your results indicate that at 84% of all days when measurements were available, stratospheric intrusions were present. This number is much higher than the previously reported numbers. For me your discussion why your number is so much higher is insufficient. Some of the differences can be explained by the good-weather bias of lidar measurements, some of them can be explained by different stratospheric intrusion detection criteria, but are these explanations enough to fully explain the huge difference in intrusion percentages? You should make sure that this discussion is focused, detailed, and thorough.

- You speculate a few times throughout the manuscript (for example in the abstract) that the change in stratospheric intrusions might be a result of improved air quality in the most relevant source regions. However, this is never really explained in detail. Please add some more information about this so that this statement is more than pure speculation.

- I would strongly recommend that the authors find a native English speaker to check the manuscript for grammar and structural problems.

Specific comments:

- Page 1, line 23: 'full tour around the northern hemisphere' -> this does not seem like a very scientific description. You might want to change this.

- Page 1, line 31: delete 'as long as'

- Page 1, line 33: delete 'of' in 'estimated of the STT. . .'

- Page 2, line 2-5: it is not clear what other characteristic than persistence is described with the sentences 'Very strong ozone. . . around the northern hemisphere (Trickl et al., 2011).' Please state them.

- Page 2, line 6: please add more specifics to '...quantify the fraction of stratospheric ozone. . .'. In the troposphere? At alpine sites?

- Page 2, line 8: 'increase' instead of 'rose'?

- Page 2, line 11: reference for 'Scheel' is missing?

- Page 2, line 14: replace 'to the beginning' with 'in 1978 at the beginning'

- Page 2, line 15: delete 'in 1978'

- Page 2, line 15: start a new sentence with 'The contribution, however, . . .'

- Page 2, line 23: state more clearly what positive trend is referred to with 'the positive trend started. . .'. Trend of stratospheric intrusions?

- Page 2, line 24: explain what kind of measurements started in 1992

- Page 2, line 25-27: sentence starting with 'For the lower-lying. . .' needs restructuring. It's meaning is not clear right now.

- Page 2, line 28: 'Scheel's approach. . .' it is not clear what that approach is, nor is it clear which reference it refers to. Please clarify.

- Page 2, line 29: replace 'this isotope' with '7Be'

- Page 2, line 29: replace 'cannot be too wrong' with 'seems plausible'

- Page 2, line 32: there is not enough information in this sentence to make its meaning unambiguous. Maybe add rephrase to 'stratospheric contribution to tropospheric ozone the overall positive tropospheric ozone trend disappears.'

- Page 2, line 33: replace 'have even expected' with 'expect'

- Page 2, line 33-34: not enough information given to understand what 'the result' means. Estimate of stratospheric intrusions?

- Page 2, line 35: 'most reasonable explanation' for what? More information needed.

- Page 2, line 35: replace 'has been a reaction of the atmospheric dynamics to the climate. . .' with 'could be a reaction of atmospheric dynamics to climate. . .'

- Page 3, line 2: 'full quantification' of what? Please specify.

- Page 3, line 5: add 'tropospheric' before 'ozone'?

- Page 3, line 6: replace 'By contrast' with 'In contrast'

- Page 3, line 32: replace 'upgrading' with 'upgrade'

- Page 4, line 8-9: wrong parenthesis for the reference, should be 'Trickl et al., (2015)'?

- Page 4, line 31: remove 'were' before 'started'

- Page 4, line 38: wrong parenthesis for references 'see also (Vogelmann...'

- Page 5, line 13: It is not clear which measurements were discontinued in January 2013. RH or ozone as well? Please be clearer.

- Page 5, line 25: replace 'routine' with 'routinely'

- Page 5, line 26: remove 'the' before ozone

- Page 5, line 27-28: replace 'in view' with 'regarding'?

- Page 5, line 31: 'to the north' not specific enough. To the north of where?

- Page 6, line 1: '1%' not specific enough. 1% of what? RH?

- Page 6, line 9: replace 'with 1°x1° horizontal resolution' with 'a 1°x1° grid'

- Page 6, line 10: 'starting in the entire region' it is not clear if that region is indeed based on the 1x1 grid?

- Page 6, line 11: how many levels are there between 250mbar and 600mbar?

- Page 6, line 19: change 'daily distributed' to 'distributed daily'

- Page 6, line 23-24: change '1x1 horizontal resolution' to '1°x1° grid'

- Page 7, line 4: add 'trajectory' before 'calculations'

- Page 7, line 24: replace 'soundings' with 'measurements'. 'Soundings' sound like they were done with ozone sondes which is not the case here, right?

- Page 7, line 26: add 'per day' after 'at least one measurement' to clarify the meaning here

- Page 7, line 32-33: the meaning of 'had gradually grown before' is not clear. Please

rephrase.

- Page 7, line 34: replace 'either for the existence of' with 'if there was'

- Page 7, line 36: replace 'looked for' with 'applied as search criteria'

- Page 7, line 37: 'had been found to be sufficient' -> sufficient for what?

- Page 7, line 38 to page 8, line 1: The sentence 'With decreasing...' does not make sense. Please rephrase.

- Page 8, line 1-2: the sentence 'At the same time...' does not make sense. Please rephrase.

- Page 8, line 2: '8-10% RH being the exception'. The meaning is not clear, please rephrase.

- Page 8, line 4: move 'in the sonde data' to after '1-2%'

- Page 8, line 5: the different types of intrusions are mentioned here, and later in the manuscript (especially Type 6). However, it is only stated that more information about the different types can be found in Trickl et al. (2010). You might want to think about explaining these different types a little more, so that readers understand what they refer to.

- Page 8, line 14: change 'Range of ozone values in intrusion layers was...' to 'Ozone values in intrusion layer ranged...'

- Page 8, line 16: remove one 'not' at the end of this line

- Page 8, line 17: replace 'looked for' with 'searched for'

- Page 8, line 26-36: It is not entirely clear why this paragraph is placed here. Does it contribute to the main message? Is it just an additional interesting case study? If it is not necessary for the main message of the manuscript, I would suggest deleting it.

- Page 9, line 5: the phrase 'the period described here' is not unambiguous. Many

different time periods have been mentioned, and at that point the reader is lost as to which this phrase refers to. Please clarify, and check the rest of the manuscript to correct the ambiguous time period references.

- Page 9, line 5: remove 'case' after 'intrusion'

- Page 9, line 6-7: combine the two sentences to 'The peak ozone mixing ratio on that day was 235 ppb, which dropped rapidly to less than 100ppb'.

- Page 9, line 9: Stop the sentence after '5ppb'. Then start a new sentence with 'Given the normally good agreement between UFS and lidar, this bias is ascribed. . .'

- Page 9, line 17: replace 'bundles' with 'calculations'?

- Page 9, line 18: 'later plots' is not specific enough. Please clarify which plots you mean.

- Page 9, line 18: replace 'South' with 'Southern'?

- Page 9, line 19: please be more specific which observations you refer to here with 'confirm the observations'

- Page 9, line 19: replace 'clearness' with 'clarity'

- Page, 9, line 20: 'separate in time and altitude' -> not clear what is meant by this

- Page 9, line 23-24: are the mentioned measurements examples for very thin layers? This is not clear from the text as it is written at the moment.

- Page 9, line 31: the mentioned example (26-27 December 2008) is not the example that is shown in Figure 4. If this is on purpose, then why not discuss the example that is shown in Figure 4 here in the text rather than discussing the 26-27 December 2008 example?

- Page 9, line 37: remove '(definition: Trickl et al., 2010)'

- Page 9, line 38: 'A source. . .' -> a source of what?

[Figure]

- Page 10, line 11: 'of' missing at the end of the line after 'upstream'?

- Page 10, line 13-20: The paragraph starting with 'Another dust case...' seems to be another example for 'intrusion layers arriving via North Africa'. Is it really necessary here for the storyline?

- Page 10, line 29: 'origin of the moisture', it is not clear what the authors mean by this. Please clarify.

- Page 10, line 30: maybe add 'during the days shown in Figure 9' after 'was reduced' for clarity

- Page 10, line 35-36: wrong parenthesis for the reference Trickl et al. (2016)

- Page 11, line 6-7: add 'e.g.' before 'Fig.8' and remove 'or more complex cases'

- Page 11, line 8: 'ones' is not unambiguous. Should this be 'intrusions' or 'trajectories' instead?

- Page 11, line 11: remove 'planetary boundary layer' and use only 'PBL' since it was already defined before

- Page 11, line 20: remove 'one to quantify easily the' with 'the easy quantification of a'

- Page 11, line 21: it is not clear what is meant with 'layer boundaries'

- Page 11, line 25: please explain how you actually calculate the measurement days with intrusion layers, if the method is only similar to the method used in Beekmann et al. (1997).

- Page 11, line 29: please add the error bars for the fraction of intrusion days to Figure 10. (this is what is referred to as 'standard deviation of 0.12', right?)

- Page 11, line 32: 'The variability is obviously much higher...' -> this is not obvious from Figure 10! Please add the error bars so that this is clear.

- Page 11, line 33: what is 'the principal seasonal cycle'? Please clarify and explain.

- Page 12, line 1: 'summer minima' of what? Stratospheric intrusions?

- Page 12, line 4: 'strong difference in seasonal cycle'. It is not clear what is meant by this. Which seasonal cycles?

- Page 12, line 5-7: please explain somewhere in this sentence what 'TT2010' means (see Figure 10).

- Page 12, line 1-15: please add the reference to the color of the bars from Figure 10 whenever they are discussed here in this section (e.g. line 8: 'lidar-based fractions', would those be the green bars?)

- Page 12, line 9: add 'to' after 'due'

- Page 12, line 23: 'the separating layer' -> it is not clear what you mean with this

- Page 12, line 24: 'initial layer thickness' -> it is not clear what you mean with this

- Page 12, line 26: add 'of weather' after 'the influence'

- Page 12, line 26: replace 'year with full coverage of all months' with ' year with measurements taken in all months'

- Page 12, line 28: 'the same high fraction', what does the 'same' refer to?

- Page 12, line 32: change 'stratosphere' to 'stratospheric'

- Page 12, line 33: did you mean 'value' instead of 'structure'? If not, it is not clear what 'structure' refers to.

- Page 12, line 36-37: change 'give rise to' to 'result in'

- Page 12, line 37: the meaning of 'discernible structure' is not clear

- Page 12, line 39: 'first years' -> which years are meant exactly by this?

- Page 13, line 2: add 'and' before 'are, thus. . .'

- Page 13, line 5: what do you mean with 'observational period'?

- Page 13, line 7: why are the results from the 'observational period' compared to the period 1996-2003? Is that period analyzed in a study? If yes, please cite which study you mean, if no, please explain why comparing the results to this period.

- Page 13, line 10-11: The two sentences starting with 'In our current effort...' do not make sense. Please rephrase or add more details for clarity.

- Page 13, line 12: 'step-like ozone rise' -> is that step like ozone rise within a profile, or on the same level with changing time? Please be more specific.

- Page 13, line 12-13: Sentence not complete? '. . .typically above 5km and contain dry layers'

- Page 13, line 15: it is not clear why the described episode is so spectacular. Please add an explanation, or avoid the very strong word spectacular.

- Page 13, line 15-25: in these two paragraphs add references to the color of the profiles that are discussed.

- Page 13, line 33: 'deviations' -> it is not clear which deviations are referred to here. Between which results?

- Page 13, line 34-36: the meaning of the sentence 'This result is in. . .' is not clear. Please rephrase.

- Page 14, line: the meaning of 'temporal coverage' here is not clear. Please explain or rephrase.

- Page 14, line 7: 'seasonal cycle peaks' -> be more specific. Seasonal cycle peaks of what?

- Page 14, line 9: which data set did Sprenger et al. (2003) analyze? Please discuss with more details!

- Page 14, line 10: 'that observational site' is not specific enough, please be more specific.

- Page 14, line 12: remove the 'the' before 'transport'

- Page 14, line 13: move 'derived' before 'seasonal cycle'

- Page 14, line 14: Change 'These pathways...' to 'However, these pathways do not always...'

- Page 14, line 15: wrong parenthesis for reference 'Forster et al., (2001)'

- Page 14, line 18: change 'call for' to 'point to'

- Page 14, line 18: what meteorological explanation would this be? Please explain.

- Page 14, line 19: 'about 15 days', is that time period different to the ∼315h of HYS-PLIT? Or does this sentence just mean that you should recalculate the earlier analyses with this ∼315h version of HYSPLIT?

- Page 14, line 25: delete 'by the observational groups in that effort'

- Page 14, line 27: replace 'asked' with 'required'

- Page 14, line 31: adjust the spelling of 'Type-6' to the spelling that is used in the rest of the manuscript 'Type 6'

- Page 14, line 33: it is not clear why the sentence starts with 'On the other hand...'. What is it the opposite to?

- Page 14, line 34-35: the sentence 'In addition, ...' is incomplete.

- Page 15, line 2: delete 'was made that'

- Page 15, line 3: meaning of the sentence 'On the other hand, just the directly detected intrusions can be used' is not clear. Please clarify.

- Page 15, line 11: 'calculated rise' -> rise of what?

[Figure]

- Page 15, line 21-22: change the sentence 'The tropopause region...' to 'The tropopause region is a mixture of about 50% stratospheric and tropospheric air each...'

- Page 15, line 23: it is not clear what you mean by 'stratospheric nature'. Please specify.

- Page 15, line 26-27: The meaning of the two sentences starting with 'Sometimes air masses...' is not clear. Please clarify.

- Figure 1, line 3: do you mean '2004' instead of '2003'?

- Figure 1, line 5: 'The stratospheric influence remarkably grew during that period' -> this is a strong statement that I cannot see very clearly in the figure. Please rephrase or explain in more detail!

- Figure 1, line 6: is the '1981' part of the sentence, or is it wrong here?

- Figures 2, 7, 9 and 12: please explain what the error bars (?) indicate in these figures! Also please add a reference to the profile of question to the figure caption or the text whenever you refer to a specific profile out of the bundle (for example: Fig 12. Intrusion layers at 3.1km (pink profile)...)

- Figure 2, line 5: one '.' too much at the end of the line.

- Figure 2, line 7: 'measurement at...' instead of 'measurements next to...'?

- Figure 3: please add a label to the color bar

- Figure 3: please add a description of what the red dot in the figure represents to the figure caption.

- Figure 4: it is not clear why this example is shown here. It is not mentioned in the text.

- Figure 10: please change the color for either 'Zugspitze fraction' or 'Zugspitze fraction (TT2010)' to something else than green. It is not clear which green bar is representing what in the figure.

- Figure 10: please explain the abbreviation 'TT2010' in the figure caption and the text

- Figure 10, line 4-5: reference 'Trickl et al., 2010' should not be fully in parenthesis.

- Figure 10 and 11: as far as I understand fractions, they cannot be greater than 1.0. Please adjust the y-axes of those two figures, since their maximum values are misleading.

- Figure 11, (first line of figure caption): you might want to add 'ozone' before 'peaks'

- Figure 12, line 3: there might be 'cycle' missing after 'diurnal'

- Figure 12: please add references to the different colored profiles to the figure caption to clarify which profile is meant when different ozone peaks are discussed.

- Figure 13: it seems like the whole figure caption is copied from Figure 8. Please provide the correct figure caption for this figure.

---

## Referee Comment (RC2) · Anonymous Referee #2 · 7 Mar 2018

The paper entitled "The underestimated role of stratosphere-to-troposphere transport on tropospheric ozone" presents an analysis of the influence of the stratosphere-to-troposphere transport on the ozone levels observed in the troposphere using the ozone and water vapor lidar database available at Garmisch-Partenkirchen from 2007 to 2016. Results obtained here are of interest for the scientific community and are very well supported by the lidar data and the ancillary information used from in-situ measurements, radiosondes and modelling tools. I recommend this paper for publication after major revisions. As a general comment, a thorough review of the language is recommended. The structure of some paragraphs and sentences is not clear at some points, complicating the understanding of the text. This is especially relevant in Section 3, where the main findings are sometimes unclear because of the writing. Besides, I

strongly suggest to include a final paragraph in Section 4 or a new section where the main conclusions of the study are clearly highlighted in a concise way. The main conclusions should be also included in the abstract in a concise statement. Sections 3 and 4 would also benefit from a more extensive and updated bibliography if possible. More detailed comments are included below:

P2, l2: "but it is not only persistence that matters." Please, add reference or develop this idea.

P2, l8: "...from 1970 to about 2003". Remove about.

P2, l29: Rephrase "the result cannot be too wrong".

P6, l9: "...interpolated to with..." Remove with. Idem for line 23.

P7, l27: Please, add Pappalardo et al. (2014).

P7, l35: Replace "has not been available" by "was not available".

P7, l36: Replace "ore" by "or". What is the descent you mention here referred to?

P8, l1: Add "were" before "preferred".

P8, l2: "This was fulfilled for most cases, 8-10% RH being the exception" Do you mean 8-10% of the cases? It is not clear.

P8, l5: Even though the reference Trickl et al (2010) is provided, the authors should provide more details about the intrusion types, especially for Type 6, which is recurrently mentioned in the text. A definition should be explicitly included.

P8, l15. Rephrase this sentence. It is not clear what the 235 ppb are referred to.

P8, l2: Rephrase "They mostly reach low altitudes above Garmisch-Partenkirchen and occur very reliably...", it is not clear.

P9, l14-22: Consider replacing LAGRANTO trajectories in Figure 3 by those obtained with HYSPLIT: According to the text here, it seems that HYSPLIT trajectories will be

more illustrative.

P9, l23-25: Check format.

P9, l33: Is it possible to add the plot of the water vapor in Fig 4?

P9, l34: Definition of Type 6 intrusions is necessary here if not included before.

P9, l37: The sentence "are shown here to become more important" is not clear.

P11, l20: Replace "to quantify easily" by "To easily quantify".

P11, l25: Can you provide here the most relevant details on the approach followed by Beekman et al. (1997)?

P12, l5: Is the fraction of days on which 3.0 km was reached the so-called Zugspitze fraction in Fig 10? Please clarify.

P12, l32-40: Additional discussion is required here. There is a large difference between the estimate based on lidar data and the estimate based on Be data. How reliable is the lidar estimate? What are possible causes for the large difference?

P13, l7: Is the period 1996-2013 is analyzed in a previous study, please provide reference and add relevant information required for the comparison with current results.

P14, l1-5: The difference between the fraction of stratospheric intrusions obtained here (84%) and previous studies is striking. A more exhaustive discussion on the possible causes is necessary. If possible increase the bibliography including additional studies for comparison. Could this difference be related to the location of the different stations and their relative position to the subtropical jet?

P14, l23-26. Do you get a better agreement if you apply these stricter criteria to your analysis?

Fig. 1. Remove 1981 from the caption.

Fig. 3. Consider replacing by HYSPLIT trajectories.

Fig. 4. Include a similar plot for the water vapor data if available.

Fig. 7. If possible, use the same vertical scale on both figures. Idem in Fig. 9.

---

## Referee Comment (RC3) · Anonymous Referee #3 · 19 Mar 2018

This paper begins with the surprising title, "The underestimated role of stratosphere-to-troposphere transport on tropospheric ozone" which implies that all previous estimates of the global STE budget over the past 20-30 years are far too low. If supported by solid evidence, this would be a landmark paper that would force a major reassessment of the global models that quantify the tropospheric ozone budget. This is an extraordinary claim and extraordinary claims require extraordinary evidence. This paper provides no such evidence as it merely reports measurements from a single location followed by speculation, incorrect assumptions and absolutely no convincing arguments that previous STE budgets are too low. In addition, the study's major conclusions were discovered and reported by other well-known papers 18-19 years earlier using more extensive datasets. I firmly recommend that this paper be rejected.

[Figure]

This review is framed around the following question: Which studies, specifically, have underestimated the role of stratosphere-to-troposphere transport on tropospheric ozone?

1) The global role of STE on the tropospheric ozone budget can only be estimated by models as the current observational network is far too limited to provide global coverage. There are many well-known studies in the literature on this topic. Stevenson et al. (2006) and Young et al. (2013) provide estimates from model ensembles, with Stevenson et al. estimating the STE global ozone flux to be approximately 550 Tg, with a range of 400-800 Tg, and Young et al. giving a slightly narrower range of 400-660 Tg. Recently, three new studies (Olsen et al., 2013; Yang et al., 2016; Jaeglé et al., 2017) have received a lot of attention from the modelling community because they estimate the STE ozone flux using the latest NASA global reanalyses that are available at much higher resolution than the chemistry-climate model simulations reported by Stevenson and Young. These reanalyses also have improved dynamics with excellent coupling between the stratosphere and the troposphere, and they assimilate satellite observations to produce accurate estimates of the stratospheric ozone burden. For example a recent paper by Knowland et al. (2017) demonstrates that the NASA MERRA-2 reanalysis accurately simulates deep stratospheric intrusions above North America due to its very high horizontal resolution of 50 km and 72 vertical layers. The average STE net ozone fluxes of the three budget studies mentioned above are 489 (Olsen), 493 (Jaeglé) and 448 Tg (Yang), similar to previous estimates reported by Stevenson and Young and similar to other estimates made in the 1990s. Essentially, the STE net ozone flux has been estimated to be roughly 500 Tg for the past 20 years, regardless of the methodology. What evidence do the authors have that all of these studies have underestimated the STE net ozone flux?

2) Despite the wide availability of publications that provide estimates of the STE net ozone flux, this paper only focuses on a single estimate of the ozone flux, a very old publication by Roelofs and Lelieveld (1997) that uses a relatively primitive global model

that was developed in 1995, with very coarse resolution of 3.75 degrees. The authors seem to approve of this model because it estimates that 40% of the tropospheric ozone burden is from the stratosphere, which agrees with the authors' estimate of stratospheric ozone above southern Germany. But the estimate by Roelofs and Lelieveld is averaged over the entire globe and across all seasons and cannot be applied to any particular location. They state that the quantity of stratospheric ozone in the troposphere varies with altitude, latitude and season. The fact that this 40% value matches the value above southern Germany is merely coincidental and does not mean that the observations from a single location can provide meaningful evaluation of an average global estimate. Even if the authors could evaluate the Roelefs and Lelieveld estimate at multiple locations and across all seasons and conclude that their estimate is correct, how would this prove that previous estimates of STE are too low? Roelofs and Lelieveld's estimate of the net global flux of stratospheric ozone into the troposphere is 459 Tg per year, and their estimate of gross photochemical ozone production is 3425 Tg per year, a factor of 7.5 greater than the net flux from the stratosphere. These numbers are entirely consistent with the ozone budget estimates described above by Stevenson, Young, Olsen, Yang and Jaegle. By accepting the results of Roelofs and Lelieveld, which agree with all the other studies, the authors are essentially agreeing with the consensus view that STE contributes roughly 450-500 TG of ozone per year to the troposphere. Again: Which studies, specifically, have underestimated the role of stratosphere-to-troposphere transport on tropospheric ozone?

3) The authors highlight the fact that their lidar detects many dry layers above southern Germany especially in summer, and claim that this is a new result. But this is not a new finding. Nearly 20 years ago the late Reginald Newell at MIT published a well-known paper in Nature in 1999 (cited 126 times according to Google Scholar) with the title: "Ubiquity of quasi-horizontal layers in the troposphere". They found, using the extensive MOZAIC database of ozone and water vapor profiles that fine layers in the atmosphere are found everywhere, and that dry layers with enhanced ozone are the most common, indicating a pervasive influence from STE. A year later V. Thouret

published an even more extensive analysis based on thousands of MOZAIC profiles around the world (half above western Europe), and here are some of their important findings:

"At northern midlatitudes we find 4 times more layers in summer than in winter, while in tropical Asia we observe a spring maximum in the occurrence of the layers. The most abundant layer type everywhere is O3+H2O− and corresponds to the signature of stratospheric intrusions or continental pollution." "The most surprising feature in this study is the lack of any seasonal variations. The only seasonal cycle and regional difference observed concern the average of the number of layers per profile (strong maximum in summer at midlatitudes and in spring in tropical Asia)." "This new finding reveals that stratospheric intrusions are not negligible in summer at midlatitudes or in the tropics, as previously thought"

Therefore, the finding by Trickl et al. regarding the high frequency of ozone-rich layers in the mid-troposphere above Europe during summer is nothing new. Thouret et al. reached the same conclusion 18 years earlier using a far more extensive data set and a much more thorough and clear analysis. The excellent and ground-breaking papers by Newell et al. and Thouret et al. demonstrate that since the publication date of these papers, STE has not been underestimated above western Europe or other regions of the northern hemisphere (where MOZAIC observations were available). Their findings were not mentioned by Trickl et al.

4) HYSPLIT is a very useful Lagrangian tool for a first look at prevailing transport patterns, but for quantitative analysis such as quantifying the amount of air or ozone transported from the stratosphere to the troposphere, it is an extremely poor choice, especially when far better models such as LAGRANTO or FLEXPART are available. The authors know how to use LAGRANTO and apply it to short-term transport analysis (<= 5 days) but for some reason, they don't use this superior tool for longer time periods. Instead they simply run HYSPLIT 15-day single back trajectories that don't give any information as to when, or if, the air parcel crossed the tropopause. They also don't give

any information as to the various source regions of the air parcel. As has been shown over and over again by the many FLEXPART studies, single back trajectories are inferior to so-called retroplumes (based on 10,000-20,000 or more trajectories), and for 15-day periods single trajectories are essentially useless. Just because a 15-day single back trajectory from a dry air layer nearly circles the globe, doesn't mean that the dry filament persisted for 15 days. The HYSPLIT analysis in this paper did nothing to further my understanding of STE processes.

5) On page 2 line 35 the authors speculate that global climate change is responsible for a doubling of stratospheric ozone at Zugspitze (increasing from 11.3 ppb to 23.4 ppb) for the 26-year time period from 1978 to 2004, but provide no analysis to back this up. They even state that this is a reasonable explanation. To the contrary, this is not a reasonable hypothesis at all. A brief check of the latest global temperature trend provided by NASA, NOAA or the EU's Copernicus shows that the global temperature over that time period increased by 0.4 degrees C. So do the authors believe that for this relatively small increase in temperature the net global STE flux increased by 100%? What would be the mechanism? A doubling of the overturning of the Brewer-Dobson circulation? Wouldn't such an enormous change in the global circulation be clearly evident in other components of the general circulation, such as a dramatic widening of the tropics? Surely such a major change in the general circulation over such a short time period would have been extensively described in the peer-reviewed literature and would be evident in the ECMWF ERA-Interim reanalysis wind fields, the same ones used by LAGRANTO. Such an analysis has already been conducted by Skerlak et al (2014) for the period 1979-2011. They find no global increase in stratosphere-to-troposphere transport. This published finding immediately disproves the speculation of the authors in the Introduction. Furthermore, an excellent paper in Nature Geoscience by Jessica Neu (NASA JPL) shows that natural fluctuations of the stratospheric circulation due to ENSO and the Quasi-biennial Oscillation (on the order of 40%) produce tropospheric ozone changes of only 2%. Perhaps there was a shift in the regional transport patterns over Europe that increased STT over 1978-2004? Based on the results of Skerlak et

al. this doesn't seem to be the case either. But given that the authors of this paper know how to use LAGRANTO, and given that the LAGRANTO analysis of Skerlak et al. must be archived somewhere, why don't they simply take the Skerlak et al. model output and explore this possibility?

6) What is the purpose of Section 3.4 Trans-Atlantic Transport? The paper is supposed to be about the underestimated role of stratosphere-to-troposphere transport on tropospheric ozone. Yet in this section they address a completely different topic which is the transport of surface ozone from North America to Europe. This is nothing new and the peer-reviewed literature is full of studies that explore this topic, studies which are far more thorough and convincing. Here the authors use a few HYSPLIT back trajectories to speculate that ozone was transported from the US boundary layer to the free troposphere above Germany. The authors speculate that the rising trajectories could be associated with a warm conveyor belt. But they provided no analysis to confirm this transport mechanism. This isn't difficult to confirm. The authors could check archived satellite imagery. The authors could also use the RAQMS model output that is shown in Figure 14. Brad Pierce has more than 20 years of experience running model analyses for aircraft campaigns all around the world. It's easy for him to quantify the contribution of the USA or STT to ozone above Germany. Instead the authors rely on just a few single HYSPLIT back trajectories. I learned nothing new from this section, and if I want to understand how ozone is transported from the USA to Europe I will consult other studies in the literature which are for more thorough and quantitative.

7) In addition to the serious problems described above, in general the manuscript is poorly written and disorganized, and tends to jump from one idea to another in a very confusing manner.

8) Finally, Hans-Eckhart Scheel is listed as a co-author but he died nearly five years ago. While his measurements may have been included in this paper, he was not able to contribute to the writing of the paper and therefore he should not be listed as a co-author as he is unable to approve of the data interpretation or the conclusions.

References:

Jaeglé, L. et al (2017), Multiyear composite view of ozone enhancements and stratosphere-to-troposphere transport in dry intrusions of northern hemisphere extratropical cyclones. Journal of Geophysical Research, 122. https://doi.org/10.1002/2017JD027656

Knowland, K. et al. (2017). Stratospheric intrusion-influenced ozone air quality exceedances investigated in the NASA MERRA-2 reanalysis, Geophysical Research Letters, 44, 10,691–10,701. https://doi.org/10.1002/2017GL074532

Neu, J. et al. (2014) Tropospheric ozone variations governed by changes in stratospheric circulation, Nature Geo., 7, 340-344.

Newell, R.E., Thouret, V., et al. (1999). Ubiquity of quasi-horizontal layers in the troposphere. Nature, 398(6725), p.316.

Olsen, M. A., Douglass, A. R., & Kaplan, T. B. (2013). Variability of extratropical ozone stratosphere–troposphere exchange using microwave limb sounder observations. Journal of Geophysical Research: Atmospheres, 118, 1090–1099. https://doi.org/10.1029/2012JD018465

Škerlak, B., M. Sprenger, and H. Wernli [2014], A global climatology of stratosphere-troposphere exchange using the ERA-Interim data set from 1979 to 2011, Atmos Chem Phys, 14, 913–937, doi:10.5194/acp-14-913-2014.

Stevenson, D. et al. [2006]. Multimodel ensemble simulations of present-day and near-future tropospheric ozone, J. Geophys. Res. 111(D8).

Thouret, V., Cho, J.Y., Newell, R.E., Marenco, A. and Smit, H.G., 2000. General characteristics of tropospheric trace constituent layers observed in the MOZAIC program. Journal of Geophysical Research: Atmospheres, 105(D13), pp.17379-17392.

Yang, H. et al [2016], Quantifying isentropic stratosphere-troposphere exchange of

ozone, J. Geophys. Res. Atmos., 121, 3372–3387; doi:10.1002/2015JD024180

Young, P. et al. [2013] Pre-industrial to end 21st century projections of tropospheric ozone from the Atmospheric Chemistry and Climate Model Intercomparison Project (ACCMIP), Atmos Chem Phys, 13, 2063–2090; doi:10.5194/ acp-13-2063-2013

---

## Author Comment (AC1) · 28 May 2018

Question for Editor: It was suggested to remove Dr. Scheel from the list of authors since he passed away several years ago. Due to his important contribution to our work on this subject I would prefer to see him as co-author.

The full reply is uploaded in the supplemant box.

Please also note the supplement to this comment: https://www.atmos-chem-phys-discuss.net/acp-2017-1192/acp-2017-1192-AC1-supplement.pdf
* * *
[Figure]

2018.

**Supplement:**

**Replies to "The underestimated role of stratosphere-to-troposphere transport on tropospheric ozone" by Thomas Trickl et al.**

T. Trickl, May 27, 2018

**General Remarks**

We thank all the reviewers for the considerable efforts spent. In general, we are quite surprised to read so many comments. Quite obviously, the rapidly approaching deadline for the special section on December 15 prevented sufficiently careful reading. I had been very tired indeed after months of analyses and a lot of urgent work, and I fully apologize for the inconvenience this created! We thank especially Reviewer 1 for particularly carefully reading the text and an unbelievably high number of suggestions.

It is obvious to me that the title can be easily misunderstood. Still, I have not found a perfect one that avoids this misunderstanding. Despite the interesting findings for both the Zugspitze summit and the lidar measurements, I do not want to selected a title of the type "Interesting results at Garmisch-Partenkirchen". There is evidence of rather high STT fractions elsewhere and this must be a part of the story. For the time being I converted the title to a question. This is adequate since we need similar analyses for more latitudes.

The high fraction of STT in the ozone observed here has been a great surprise to us. One component of this is the positive trend of this component since 1975, which calls for more analyses. However, we expect that these finding could be valid in other regions as well. Long data series are unfortunately rare! I tried to include material from other sites or observational platforms. The recent analyses in the U.S. (Table Mountain, Huntsville) are highly valuable in this regard. I rewrote the Introduction to make our argumentation clearer.

In the following, we insert the text from the reports in italics and our replies in normal text.

**Anonymous Referee #1**

 The manuscript "The underestimated role of stratosphere-to-troposphere transport on tropospheric ozone" by Trickl et al. describes an analysis of the effect of stratospheric intrusions over Garmisch-Partenkirchen on ozone values over the full free troposphere. Long-term ozone and water vapor lidar measurements from Garmisch-Partenkirchen, in-situ and radiosonde data from nearby stations, and trajectory analyses combined provide a very good data basis for these kinds of analyses. Different intrusion events are categorized, and possible reasons for changes in event frequencies are mentioned. The structure of the manuscript is somewhat chaotic so that it is difficult to find the storyline in the text. Sometimes the sentence structures are not entirely correct which complicates the understanding of the meaning of phrases and sentences. I believe the applied analyses methods are sound, however, it is hard to tell since there is an abundance of information about historical analyses already published, recent analyses, and planned analyses. I think this manuscript needs major restructuring, rewording and general clarifications before the full extent of the presented analyses becomes clear. But I do believe that the study can show some interesting results if they are presented in a precise and detailed manner. I can only recommend the manuscript for publication if major revisions have been done, following the general and detailed suggestions listed below.*

We do understand what is pointed out here. We reorganized and extended several parts of the paper, in particular the Introduction and the Discussion sections. We see our results as a highlight of a development during the past two decades during which the contribution of STT to tropospheric has strongly increased from estimates of less than 10 ppb. We now start from the strong increase of

background ozone with growing air pollution, but find that this increase has hidden a rise of STT in our area.

**General suggestions/comments:**

*- The manuscript has both too much and not enough information in parts. Very often, much background information about measurements, specific intrusion cases etc. are given that might not be necessary for the storyline (some of these will be pointed out in the specific comment section). And then there are crucial information parts missing to be able to understand which exact intrusion case is actually referred to, or which reference is referred to (some of these will be pointed out in the specific comment section). I strongly recommend checking the whole manuscript for these things to enhance the clarity of the analysis and the storyline. At the moment the storyline (what was exactly done, why was it done, how was it done) is not obvious, which makes reading the manuscript and understanding its message very difficult.*

Done.

*- The manuscript would be easier to read if the authors would refer to specific references, measurement systems or intrusion cases in the same way. For example on page 14, the reference 'Beekmann et al. (1997)' is discussed. However, within this paragraph, this study is also referred to as 'the 1997 analysis' or 'the 1997 results'. Repeating the reference as 'Beekmann et al. (1997)' is not an unnecessary repetition of words, but actually very helpful in understanding what is discussed in the paragraph. Please go through the manuscript and standardize the description of the same reference/ results/methods.*

Done

*- Very often just one or two words are missing to clarify the meaning of a sentence or a phrase. For example, on page 14, line 7, 'seasonal cycle peaks' is not unambiguous. 'Season cycle peaks' of what exactly? Some more of the ambiguous descriptions will be provided in the special comments section, but I encourage the authors strongly to check the manuscript independently for these ambiguous descriptions.*

I changed a lot prior to reading the comments in detail. Also Dr. Sprenger made helpful adjustments.

*- Abstract: The first two sentences are more relevant to an introduction than an abstract. This might be the motivation for the study, but does nothing to explain exactly what has been done in the analysis. Please rewrite the abstract to include all the important parts of information necessary to understand the performed analysis (for example, the exact time period on which the analysis is based), and also state very clearly the main findings of the study.*

Thank you for pointing this out! The abstract was derived from the conference abstract. Under pressure to meet the deadline quite obviously adjustments to the changing style of the manuscript were forgotten.

*- The main message of the manuscript is buried in many details and case studies. I think it is necessary to restructure the manuscript that the results of the main analyses become clearer. You present systematic analysis of stratospheric intrusions over the whole troposphere; you discover a change in the annual cycle of intrusion frequencies when you do this; there are many more intrusion layers than previously reported (did I understand this correctly?); some of the changes might be due to a change in climate. If these are the main points of your study, make sure that every reader will understand this and the connections between these points. – The use of 7Be as tracer for descending air is explained in detail at the beginning of the manuscript. However, in the discussion section it is mentioned that analyses including 7Be are only planned for the future and are not done yet. Why the lengthy explanation about this method then earlier in the document?*

We do not want to present case studies. The idea is a collection of typical findings during the decade of measurements considered. The methods are described in the section of methods. The question about the obvious increase of STT in the Zugspitze data cannot be solved for the lidar data (we average over the full ten years in order to obtain reasonable statistics anyway!). The STT trend in Zugspitze $O_3$ will be analysed in the follow-up paper on a full re-analysis. This had to be postponed due to Dr. Scheel´s death. I have planned for many years to look at changing intrusion frequencies or the ozone content of the layers.

Mentioning [7]Be is important: This is the great advantage of the in-situ data in the trend analysis and for estimating the indirect contribution (here, vertical sounding has its limits!). In addition, the detection of the doubling of STT at this site is an important finding that underlines the title. Nevertheless, we shortened the description of the in-situ results.

*- Your results indicate that at 84% of all days when measurements were available, stratospheric intrusions were present. This number is much higher than the previously reported numbers. For me your discussion why your number is so much higher is insufficient. Some of the differences can be explained by the good-weather bias of lidar measurements, some of them can be explained by different stratospheric intrusion detection criteria, but are these explanations enough to fully explain the huge difference in intrusion percentages? You should make sure that this discussion is focused, detailed, and thorough.*

I had contacted Dr. Beekmann for discussing this issue, but have not received a reply. His paper clearly uses more restrictive criteria. I tried to mention the differences, but a quantitative assessment of their impact would require a funded project, considering the many months of analysis required for the current study.

*- You speculate a few times throughout the manuscript (for example in the abstract) that the change in stratospheric intrusions might be a result of improved air quality in the most relevant source regions. However, this is never really explained in detail. Please add some more information about this so that this statement is more than pure speculation.*

A change in STT is not ascribed to improved air quality. The Zugspitze data show increasing STT which explains the increasing ozone despite the reported improving air quality.

*- I would strongly recommend that the authors find a native English speaker to check the manuscript for grammar and structural problems.*

**Specific comments:**

This really a great review, perhaps the longest I have ever received. I am sorry for so many mistakes!

*- Page 1, line 23: 'full tour around the northern hemisphere' -> this does not seem like a very scientific description. You might want to change this.*

Changed.

*- Page 1, line 31: delete 'as long as'*

Deleted.

*- Page 1, line 33: delete 'of' in 'estimated of the STT: : :'*

Deleted.

*- Page 2, line 2-5: it is not clear what other characteristic than persistence is described with the sentences 'Very strong ozone: : : around the northern hemisphere (Trickl et al., 2011).' Please state them.*

I do not understand this suggestion: We cannot state all six cases and their dates! The issue here is "high ozone", not the persistence. I modified the paragraph.

*- Page 2, line 6: please add more specifics to '...quantify the fraction of stratospheric ozone: : :'. In the troposphere? At alpine sites?*

I changed the entire paragraph. Thus, this suggestion no longer matters.

*- Page 2, line 8: 'increase' instead of 'rose'?*

Changed.

*- Page 2, line 11: reference for 'Scheel' is missing?*

It is not missing: it is cited at the end of the sentence! I moved the reference to "Scheel" for more clearness.

*- Page 2, line 14: replace 'to the beginning' with 'in 1978 at the beginning'*

No longer present in shortened new version of the description.

*- Page 2, line 15: delete 'in 1978'*

No longer present in shortened new version of the description.

*- Page 2, line 15: start a new sentence with 'The contribution, however, : : :'*

Changed.

*- Page 2, line 23: state more clearly what positive trend is referred to with 'the positive trend started: : :'. Trend of stratospheric intrusions?*

Changed.

*- Page 2, line 24: explain what kind of measurements started in 1992*

Changed.

*- Page 2, line 25-27: sentence starting with 'For the lower-lying: : :' needs restructuring. It's meaning is not clear right now.*

The second part of the sentence was rewriten.

*- Page 2, line 28: 'Scheel's approach: : :' it is not clear what that approach is, nor is it clear which reference it refers to. Please clarify.*

Clarified.

*- Page 2, line 29: replace 'this isotope' with '7Be'*

Replaced.

*- Page 2, line 29: replace 'cannot be too wrong' with 'seems plausible'*

Replaced.

*- Page 2, line 32: there is not enough information in this sentence to make its meaning unambiguous. Maybe add rephrase to 'stratospheric contribution to tropospheric ozone the overall positive tropospheric ozone trend disappears.'*

I now refer to Fig. 1.

*- Page 2, line 33: replace 'have even expected' with 'expect'*

Changed.

*- Page 2, line 33-34: not enough information given to understand what 'the result' means. Estimate of stratospheric intrusions?*

New: Thus, the estimate of the stratospheric component is perhaps even somewhat conservative.

*- Page 2, line 35: 'most reasonable explanation' for what? More information needed.*

Added.

*- Page 2, line 35: replace 'has been a reaction of the atmospheric dynamics to the climate: : :' with 'could be a reaction of atmospheric dynamics to climate: : :'*

Partly changed. However, "climate change" looks clearer to me.

*- Page 3, line 2: 'full quantification' of what? Please specify.*

I removed the entire paragraph

*- Page 3, line 5: add 'tropospheric' before 'ozone'?*

New: The seasonal cycle of the stratospheric ozone contribution

*- Page 3, line 6: replace 'By contrast' with 'In contrast'*

New: In contrast to this.

*- Page 3, line 32: replace 'upgrading' with 'upgrade'*

Done.

*- Page 4, line 8-9: wrong parenthesis for the reference, should be 'Trickl et al., (2015)'?*

To my knowledge the parentheses are correct.

*- Page 4, line 31: remove 'were' before 'started'*

Removed.

*- Page 4, line 38: wrong parenthesis for references 'see also (Vogelmann: : :'*

Comma inserted.

*- Page 5, line 13: It is not clear which measurements were discontinued in January 2013. RH or ozone as well? Please be clearer.*

Should be clear from the content of this chapter! I inserted "in-situ".

*- Page 5, line 25: replace 'routine' with 'routinely'*

Gracious Lord! Changed!

*- Page 5, line 26: remove 'the' before ozone*

No: the ozone number density!

*- Page 5, line 27-28: replace 'in view' with 'regarding'?*

Changed.

*- Page 5, line 31: 'to the north' not specific enough. To the north of where?*

I added "of IFU"

*- Page 6, line 1: '1%' not specific enough. 1% of what? RH?*

"RH" added.

*- Page 6, line 9: replace 'with 1_x1_ horizontal resolution' with 'a 1_x1_ grid'*

Changed.

*- Page 6, line 10: 'starting in the entire region' it is not clear if that region is indeed based on the 1x1 grid?*

It is! In (Trickl et al., 2010) H. Feldmann made a single-year Eulerian run with this grid size.

*- Page 6, line 11: how many levels are there between 250mbar and 600mbar?*

Added.

*- Page 6, line 19: change 'daily distributed' to 'distributed daily'*

Changed.

*- Page 6, line 23-24: change '1x1 horizontal resolution' to '1_x1_ grid'*

Changed.

*- Page 7, line 4: add 'trajectory' before 'calculations'*

Added.

*- Page 7, line 24: replace 'soundings' with 'measurements'. 'Soundings' sound like they were done with ozone sondes which is not the case here, right?*

Vertical sounding is mostly used in a more general way. Nevertheless, I changed the expression to "ozone profiles".

*- Page 7, line 26: add 'per day' after 'at least one measurement' to clarify the meaning here*

I do not agree! In all sentences we talk about single days, also is this one. Thus, the meaning is obvious. Here, the climatology days are referred to in the very same sentence.

*- Page 7, line 32-33: the meaning of 'had gradually grown before' is not clear. Please rephrase.*

I use "improved" now.

*- Page 7, line 34: replace 'either for the existence of' with 'if there was'*

Good suggestion, adopted!

*- Page 7, line 36: replace 'looked for' with 'applied as search criteria'*

Changed to: applied as criterion to verify a stratospheric layer.

*- Page 7, line 37: 'had been found to be sufficient' -> sufficient for what?*

I added: for identifying STT

*- Page 7, line 38 to page 8, line 1: The sentence 'With decreasing: : :' does not make sense. Please rephrase.*

Changed to: At lower latitudes

*- Page 8, line 1-2: the sentence 'At the same time: : :' does not make sense. Please rephrase.*

New: In addition, ....

*- Page 8, line 2: '8-10% RH being the exception'. The meaning is not clear, please rephrase.*

I deleted the entire phrase.

*- Page 8, line 4: move 'in the sonde data' to after '1-2%'*

Done.

*- Page 8, line 5: the different types of intrusions are mentioned here, and later in the manuscript (especially Type 6). However, it is only stated that more information about the different types can be found in Trickl et al. (2010). You might want to think about explaining these different types a little more, so that readers understand what they refer to.*

I now write: "mostly by distinguishing source regions". In the case of Types 1 and 2 anti-cyclonic and cyclonic advection from Greenland and Iceland to Garmisch-Partenkirchen is distinguished These types are not explicitly discussed in the paper and, thus, I omit an explanation in order to reduce complexity.

*- Page 8, line 14: change 'Range of ozone values in intrusion layers was: : :' to 'Ozone values in intrusion layer ranged: : :'*

I already removed the entire paragraph since this information is found below. Much better style!

*- Page 8, line 16: remove one 'not' at the end of this line*

Same.

*- Page 8, line 17: replace 'looked for' with 'searched for'*

Same.

*- Page 8, line 26-36: It is not entirely clear why this paragraph is placed here. Does it contribute to the main message? Is it just an additional interesting case study? If it is not necessary for the main message of the manuscript, I would suggest deleting it.*

I shortened this paragraph. However, to my opinion it is helpful as an overview.

*- Page 9, line 5: the phrase 'the period described here' is not unambiguous. Many different time periods have been mentioned, and at that point the reader is lost as to which this phrase refers to. Please clarify, and check the rest of the manuscript to correct the ambiguous time period references.*

Clarified.

*- Page 9, line 5: remove 'case' after 'intrusion'*

Obvious! Done.

*- Page 9, line 6-7: combine the two sentences to 'The peak ozone mixing ratio on that day was 235 ppb, which dropped rapidly to less than 100ppb'.*

Good suggestion! Done.

*- Page 9, line 9: Stop the sentence after '5ppb'. Then start a new sentence with 'Given the normally good agreement between UFS and lidar, this bias is ascribed: : :'*

Done.

*- Page 9, line 17: replace 'bundles' with 'calculations'?*

No! New intrusion branches show up that add complexity. I re-organized this part.

*- Page 9, line 18: 'later plots' is not specific enough. Please clarify which plots you mean.*

Done.

*- Page 9, line 18: replace 'South' with 'Southern'?*

I had a look into my Dictionary: Southern is correct. Interesting after that may years of publishing !

*- Page 9, line 19: please be more specific which observations you refer to here with 'confirm the observations'*

"in Fig. 2" added.

*- Page 9, line 19: replace 'clearness' with 'clarity'*

Thank you for this suggestion! Done.

*- Page, 9, line 20: 'separate in time and altitude' -> not clear what is meant by this*

I modified this sentence and the next one.

*- Page 9, line 23-24: are the mentioned measurements examples for very thin layers? This is not clear from the text as it is written at the moment.*

For unknown reasons the sentence starting with "235 ppb" merged with the following paragraph; this was corrected. The examples mentioned are, indeed, for thin layers. I changed the two sentences to:

"A particularly spectacular case (26-27 December 2008) of a thin layer was discussed by Trickl et al. (2014) and verified by high-vertical-resolution transport modelling. Here, we show as an example an even more exciting case of two parallel very thin high-ozone layers descending to Alpine summit levels (Fig. 4)."

*- Page 9, line 31: the mentioned example (26-27 December 2008) is not the example that is shown in Figure 4. If this is on purpose, then why not discuss the example that is shown in Figure 4 here in the text rather than discussing the 26-27 December 2008 example?*

Both examples are from the period 2007-2016. The first one is already published and cannot be discussed again here. The December-2013 case is even more spectacular. I added the sentence "Again, the minimum RH was 1 %, the cut-off level of the sonde (Trickl et al., 2014)."

*- Page 9, line 37: remove '(definition: Trickl et al., 2010)'*

Done (indeed already introduced)

*- Page 9, line 38: 'A source: : :' -> a source of what?*

"of STT" added.

*- Page 10, line 11: 'of' missing at the end of the line after 'upstream'?*

I looked for examples: "of" is more frequent.

*- Page 10, line 13-20: The paragraph starting with 'Another dust case: : :' seems to be another example for 'intrusion layers arriving via North Africa'. Is it really necessary here for the storyline?*

The first case could, indeed, be enough, but there is no lidar measurement of water vapour. In the second case the intrusion does not pass over North Africa. However, in this example where we have water vapour from the lidar. We need such an example! The example at least shows that the pathways can be different.

*- Page 10, line 29: 'origin of the moisture', it is not clear what the authors mean by this. Please clarify.*

I added "required for the cloud formation."

*- Page 10, line 30: maybe add 'during the days shown in Figure 9' after 'was reduced' for clarity*

New: As indicated by the error bars in Fig. 9 the quality of the lidar measurements in the upper troposphere is reduced in summer due to the strong absorption of the laser radiation by the elevated ozone and due to the high level of scattered sunlight. The vertical range is slightly larger during the darker hours.

*- Page 10, line 35-36: wrong parenthesis for the reference Trickl et al. (2016)*

This way of citing looks more logical to me. It is usually changed by the journals, and I expect the same to happen here.

*- Page 11, line 6-7: add 'e.g.' before 'Fig.8' and remove 'or more complex cases'*

Changed.

*- Page 11, line 8: 'ones' is not unambiguous. Should this be 'intrusions' or 'trajectories' instead?*

Changed to: Only these verified cases were accepted as "stratospheric", given the low humidity.

*- Page 11, line 11: remove 'planetary boundary layer' and use only 'PBL' since it was*

*already defined before*

Changed.

*- Page 11, line 20: remove 'one to quantify easily the' with 'the easy quantification of a'*

Changed.

*- Page 11, line 21: it is not clear what is meant with 'layer boundaries'*

I do not fully understand this remark. I modified the sentence for more clarity: The boundaries of the intrusion layers cannot always be clearly distinguished.

*- Page 11, line 25: please explain how you actually calculate the measurement days with intrusion layers, if the method is only similar to the method used in Beekmann et al. (1997).*

Do you mean "define the measurement days"? Or do you mean "calculate the number of measurement days"? I added "per month" since this was missing.

*- Page 11, line 29: please add the error bars for the fraction of intrusion days to Figure 10. (this is what is referred to as 'standard deviation of 0.12', right?)*

The error bars are uncertain. As mentioned the standard deviation of 0.12 is not realistic. Thus, I prefer to stay just with a comment.

*- Page 11, line 32: 'The variability is obviously much higher: : :' -> this is not obvious from Figure 10! Please add the error bars so that this is clear.*

I added: "As seen from a comparison with the fractions derived by including all measurements days,". The excursions of the values in January, February and November is obvious.

*- Page 11, line 33: what is 'the principal seasonal cycle'? Please clarify and explain.*

Chnaged to: Nevertheless, the principal course of the seasonal cycle is retained

*- Page 12, line 1: 'summer minima' of what? Stratospheric intrusions?*

I added "of deep STT".

*- Page 12, line 4: 'strong difference in seasonal cycle'. It is not clear what is meant by this. Which seasonal cycles?*

Changed to: there must be a pronounced free-tropospheric summer maximum of STT to explain the overall seasonal cycle with a slight maximum in August

*- Page 12, line 5-7: please explain somewhere in this sentence what 'TT2010' means (see Figure 10).*

It is now explained in the figure caption.

*- Page 12, line 1-15: please add the reference to the color of the bars from Figure 10 whenever they are discussed here in this section (e.g. line 8: 'lidar-based fractions', would those be the green bars?)*

New figure caption: **Fig. 10.** Fraction of intrusion days in the ozone lidar data averaged for each month between 2007 and 2016 (see text); we give the fraction for all measurement days (red bars) and for the "climatology days" Monday and Thursday (transparent bars). For comparison, the same analysis is shown for those intrusion days that show stratospheric influence at 2962 m (Zugspitze, dark green), together with the maximum fractions calculated from the Zugspitze in-situ data underlying Fig. 12 in (Trickl et al., 2010, "TT2010"; light green).

*- Page 12, line 9: add 'to' after 'due'*

Inserted.

*- Page 12, line 23: 'the separating layer' -> it is not clear what you mean with this*

New: intrusion layer extends into the stratosphere when the descent starts.

*- Page 12, line 24: 'initial layer thickness' -> it is not clear what you mean with this*

See above!

*- Page 12, line 26: add 'of weather' after 'the influence'*

This not the motivation! I was astonished when reading the text and changed it completely. 2014 was the year with the with the highest coverage with data and, therefore, the best year for estimating the stratospheric impact.

*- Page 12, line 26: replace 'year with full coverage of all months' with ' year with measurements taken in all months'*

Changed.

*- Page 12, line 28: 'the same high fraction', what does the 'same' refer to?*

I added "more than 80" for clarification.

*- Page 12, line 32: change 'stratosphere' to 'stratospheric'*

Changed.

*- Page 12, line 33: did you mean 'value' instead of 'structure'? If not, it is not clear what 'structure' refers to.*

I completely modified this part for more clarity.

*- Page 12, line 36-37: change 'give rise to' to 'result in'*

I prefer to use "exhibit".

*- Page 12, line 37: the meaning of 'discernible structure' is not clear*

Changed to "discernible ozone peaks"

*- Page 12, line 39: 'first years' -> which years are meant exactly by this?*

Fig. 1 end with the year 2004. Thus, the meaning is obvious.

*- Page 13, line 2: add 'and' before 'are, thus: : :'*

Inserted.

*- Page 13, line 5: what do you mean with 'observational period'?*

The first half of this section was completely rewritten. The motivation is on questions like: Can we identify elevated ozone from intercontinental transport in the presence of so many STT layers? Many ranges of elevated ozone are not dry, but also cannot attributed to remote PBLs.

*- Page 13, line 7: why are the results from the 'observational period' compared to the period 1996-2003? Is that period analyzed in a study? If yes, please cite which study you mean, if no, please explain why comparing the results to this period.*

I made a FLEXTRA trajectory study for 2003-2005 (published just in a report) showing that we have at least 28 % of areas in the U.S. with strong air pollution; from this elevated fraction one would expect more elevated U.S. ozone in our data.

I changed this part anyway.

*- Page 13, line 10-11: The two sentences starting with 'In our current effort...' do not make sense. Please rephrase or add more details for clarity.*

This no longer exist after rewriting this part.

*- Page 13, line 12: 'step-like ozone rise' -> is that step like ozone rise within a profile, or on the same level with changing time? Please be more specific.*

Referring to Fig. 9 should be enough! However, I added "in the profiles"

*- Page 13, line 12-13: Sentence not complete? ': : :typically above 5km and contain dry layers'*

I already split this into two sentences for clarity.

*- Page 13, line 15: it is not clear why the described episode is so spectacular. Please add an explanation, or avoid the very strong word spectacular.*

I replaced "spectacular ozone" by "high-ozone".

*- Page 13, line 15-25: in these two paragraphs add references to the color of the profiles that are discussed.*

Done.

- Page 13, line 33: 'deviations' -> it is not clear which deviations are referred to here. Between which results?

I added "from Fig. 13".

*- Page 13, line 34-36: the meaning of the sentence 'This result is in: : :' is not clear. Please rephrase.*

Split into two sentences.

*- Page 14, line: the meaning of 'temporal coverage' here is not clear. Please explain or rephrase.*

I added "of the measurements".

*- Page 14, line 7: 'seasonal cycle peaks' -> be more specific. Seasonal cycle peaks of what?*

Inserted!

*- Page 14, line 9: which data set did Sprenger et al. (2003) analyze? Please discuss with more details!*

I added "with a LAGRANTO trajectory analysis".

*- Page 14, line 10: 'that observational site' is not specific enough, please be more specific.*

In this part of the paragraph on one observational site is mentioned! I changed the entire end of the paragraph, also adding one more very recent reference supporting the high start stratospheric fractions.

*- Page 14, line 12: remove the 'the' before 'transport'*

Deleted.

*- Page 14, line 13: move 'derived' before 'seasonal cycle'*

Shifted.

*- Page 14, line 14: Change 'These pathways: : :' to 'However, these pathways do not always: : :'*

Changed.

*- Page 14, line 15: wrong parenthesis for reference 'Forster et al., (2001)'*

Corrected.

*- Page 14, line 18: change 'call for' to 'point to'*

This is not what we mean. A meteorological explanation is needed, I can point to it only if already exists.

*- Page 14, line 18: what meteorological explanation would this be? Please explain.*

I do not know! However, it would interesting to see if there is some reproducible pattern in this latitudinal range.

*- Page 14, line 19: 'about 15 days', is that time period different to the _315h of HYSPLIT? Or does this sentence just mean that you should recalculate the earlier analyses with this _315h version of HYSPLIT?*

Sometimes 315 h is not sufficient and the trajectories indicate additional rise at the end. 15 days is the longest descent seen. I added "(as suggested by our analyses)".

*- Page 14, line 25: delete 'by the observational groups in that effort'*

No! There is a second assessment in than needed for distinguishing!

*- Page 14, line 27: replace 'asked' with 'required'*

Changed!

*- Page 14, line 31: adjust the spelling of 'Type-6' to the spelling that is used in the rest of the manuscript 'Type 6'*

Spelling checked and changes made where necessary.

*- Page 14, line 33: it is not clear why the sentence starts with 'On the other hand: : :'. What is it the opposite to?*

I deleted the entire sentence since it could be confusing.

*- Page 14, line 34-35: the sentence 'In addition, : : :' is incomplete.*

Unbelievable! I added "must be considered"

*- Page 15, line 2: delete 'was made that'*

Deleted. Good suggestion!

*- Page 15, line 3: meaning of the sentence 'On the other hand, just the directly detected intrusions can be used' is not clear. Please clarify.*

Changed.

*- Page 15, line 11: 'calculated rise' -> rise of what?*

Changed to "ozone rise"

*- Page 15, line 21-22: change the sentence 'The tropopause region: : :' to 'The tropopause region is a mixture of about 50% stratospheric and tropospheric air each: : :'*

Changed.

*- Page 15, line 23: it is not clear what you mean by 'stratospheric nature'. Please specify.*

Change to "portion".

*- Page 15, line 26-27: The meaning of the two sentences starting with 'Sometimes air masses: : :' is not clear. Please clarify.*

I added a colon before this sentence. It is related the previous one.

*- Figure 1, line 3: do you mean '2004' instead of '2003'?*

You are right: Changed! The ATMOFAST final report was submitted in 2005, and the data selection, therefore, ended one year earlier.

*- Figure 1, line 5: 'The stratospheric influence remarkably grew during that period' -> this is a strong statement that I cannot see very clearly in the figure. Please rephrase or explain in more detail!*

New: doubled.

*- Figure 1, line 6: is the '1981' part of the sentence, or is it wrong here?*

I added "after"

*- Figures 2, 7, 9 and 12: please explain what the error bars (?) indicate in these figures! Also please add a reference to the profile of question to the figure caption or the text whenever you refer to a specific profile out of the bundle (for example: Fig 12. Intrusion layers at 3.1km (pink profile): : :)*

I added two more error bars and a few explanations to Fig. 2 since this series is rather tutorial. I think there is no need to repeat all this in the other captions, in agreement with what one most commonly finds in the literature.

*- Figure 2, line 5: one '.' too much at the end of the line.*

Deleted.

*- Figure 2, line 7: 'measurement at: : :' instead of 'measurements next to: : :'?*

No: The time grid of the in-situ data is 0.5 h.

*- Figure 3: please add a label to the color bar*

This is reasonable, although the scale is described in the caption! Added.

*- Figure 3: please add a description of what the red dot in the figure represents to the figure caption.*

I thought I had added such a description! Now it is there!

*- Figure 4: it is not clear why this example is shown here. It is not mentioned in the text.*

It **is** mentioned in the text (item 2 in Sec. 3.2)!!! Figs. 2 and 4 are presented in order to give examples for the extremes than occur in the data. I also added a sentence on the minimum sonde

RH: Again, it is 1 % which is the minimum of the RS92 scale, e.g., it could also be 0 %! This is remarkable for a thin layer!

*- Figure 10: please change the color for either 'Zugspitze fraction' or 'Zugspitze fraction (TT2010)' to something else than green. It is not clear which green bar is representing what in the figure.*

I do not agree! On my screen there is excellent contrast of both colours, and I selected clearly different width for the two quantities. I had tried other colours, but green does not fit well to many colours.

*- Figure 10: please explain the abbreviation 'TT2010' in the figure caption and the text*

Now explained.

*- Figure 10, line 4-5: reference 'Trickl et al., 2010' should not be fully in parenthesis.*

I do not agree. This should be decided by the proof editor according to the journal style.

*- Figure 10 and 11: as far as I understand fractions, they cannot be greater than 1.0.*

This is obviously not the case! In any case, the frame must be longer in order to provide space for the legend. I retained the size of the frame, but ended the scale at 1.0.

*Please adjust the y-axes of those two figures, since their maximum values are misleading.*

See above.

*- Figure 11, (first line of figure caption): you might want to add 'ozone' before 'peaks'*

Inserted.

*- Figure 12, line 3: there might be 'cycle' missing after 'diurnal'*

Inserted.

*- Figure 12: please add references to the different colored profiles to the figure caption to clarify which profile is meant when different ozone peaks are discussed.*

This is specified in the legends! The letter size of the legends was chosen to be readable after size reduction.

*- Figure 13: it seems like the whole figure caption is copied from Figure 8. Please provide the correct figure caption for this figure.*

I modified the caption.

**I owe you a bottle of wine!!!**

**Anonymous Referee #2**

*The paper entitled "The underestimated role of stratosphere-to-troposphere transport on tropospheric ozone" presents an analysis of the influence of the stratosphereto- troposphere transport on the ozone levels observed in the troposphere using the ozone and water vapor lidar database available at Garmisch-Partenkirchen from 2007 to 2016. Results obtained here are of interest for the scientific community and are very well supported by the lidar data and the ancillary information used from in-situ measurements, radiosondes and modelling tools. I recommend this paper for publication after major revisions. As a general comment, a thorough review of the language is recommended. The structure of some paragraphs and sentences is not clear at some*

*points, complicating the understanding of the text. This is especially relevant in Section 3, where the main findings are sometimes unclear because of the writing. Besides, I strongly suggest to include a final paragraph in Section 4 or a new section where the main conclusions of the study are clearly highlighted in a concise way. The main conclusions should be also included in the abstract in a concise statement. Sections 3 and 4 would also benefit from a more extensive and updated bibliography if possible.*

I fully agree! Reading the text again after quite a few months I realized that several parts (in particular the Introduction) are not organized in the way I wanted to tell the story. The Introduction is now written to make clear what the title mostly means: It is not all photochemical production! Our conclusions are based also on results from other information. A few more references were found and inserted. More conclusions were added in Sec. 4.

*More detailed comments are included below:*

*P2, l2: "but it is not only persistence that matters." Please, add reference or develop this idea.*

The reference is cited in the preceding sentence. However, having a closer look I found that (Sprenger et al., 2003) is not the best source. Koch et al. (2006; same group) gives more information.,

*P2, l8: ". . .from 1970 to about 2003". Remove about.*

Removed. This part was shifted downward in the text.

*P2, l29: Rephrase "the result cannot be too wrong".*

Changed as suggested by Rev. 1.

*P6, l9: ". . .interpolated to with. . ." Remove with. Idem for line 23.*

Changed.

*P7, l27: Please, add Pappalardo et al. (2014).*

This paper just mentions a "regular schedule" without specifying it.

*P7, l35: Replace "has not been available" by "was not available".*

Changed.

*P7, l36: Replace "ore" by "or". What is the descent you mention here referred to?*

I added "of the trajectories" and "applied as criterion to verify a stratospheric layer".

*P8, l1: Add "were" before "preferred".*

I added "have been" since this kind of analysis is still done.

*P8, l2: "This was fulfilled for most cases, 8-10% RH being the exception" Do you mean 8-10% of the cases? It is not clear.*

Superfluous, deleted.

*P8, l5: Even though the reference Trickl et al (2010) is provided, the authors should provide more details about the intrusion types, especially for Type 6, which is recurrently mentioned in the text. A definition should be explicitly included.*

Types 1-5 are not relevant here. Type 6 is defined. I added "mostly by distinguishing source regions".

*P8, l15. Rephrase this sentence. It is not clear what the 235 ppb are referred to.*

This paragraph was deleted since the details are given in the following.

*P8, l2: Rephrase "They mostly reach low altitudes above Garmisch-Partenkirchen and occur very reliably...", it is not clear.*

I added "after virtually all cold fronts".

*P9, l14-22: Consider replacing LAGRANTO trajectories in Figure 3 by those obtained with HYSPLIT: According to the text here, it seems that HYSPLIT trajectories will be more illustrative.*

This had been considered. But I need at least one LAGRANTO example! In the other examples longer transport times than 4-5 days are involved.

*P9, l23-25: Check format.*

This had accidentally occurred for unknown reason and was already corrected. The first sentence belongs to the previous paragraph!

*P9, l33: Is it possible to add the plot of the water vapor in Fig 4?*

I do not have such a nice time series form water vapour since UFS is closed around the end of the year which prevents lidar measurements. A statement about humidity is important, and I already added a sentence verifying minimum RH = 1 % in these thin layers, as also found in other thin intrusion layers.

*P9, l34: Definition of Type 6 intrusions is necessary here if not included before.*

It is defined before!

*P9, l37: The sentence "are shown here to become more important" is not clear.*

Changed to "were observed much more frequently above 4.5 km than at the Zugspitze summit."

*P11, l20: Replace "to quantify easily" by "To easily quantify".*

This would be a split infinitive that is forbidden by the English grammar.

*P11, l25: Can you provide here the most relevant details on the approach followed by Beekman et al. (1997)?*

This is done in the Discussion. I now mention this here. Here, just the fact is needed that Beekmann et al. also determined the temporal coverage.

*P12, l5: Is the fraction of days on which 3.0 km was reached the so-called Zugspitze fraction in Fig 10? Please clarify.*

Clarified in the following sentence.

*P12, l32-40: Additional discussion is required here. There is a large difference between the estimate based on lidar data and the estimate based on Be data. How reliable is the lidar estimate? What are possible causes for the large difference?*

I think the difference is not that large. We explicitly mention the limited data coverage for the lidar measurements, whereas as in-situ data are available around the clock.

*P13, l7: Is the period 1996-2013 is analyzed in a previous study, please provide reference and add relevant information required for the comparison with current results.*

*P14, l1-5: The difference between the fraction of stratospheric intrusions obtained here (84%) and previous studies is striking. A more exhaustive discussion on the possible causes is necessary. If possible increase the bibliography including additional studies for comparison. Could this difference be related to the location of the different stations and their relative position to the subtropical jet?*

No, not to this extent! I do not know similar studies. I contacted Dr. Beekmann for his opinion, but I did not receive any reply. I describe pretty much what I found in that paper. Perhaps their description of the filtering approach is not complete. For us it is more important for the time being that two stations in the U.S. found stratospheric fraction of 50 % and more.

*P14, l23-26. Do you get a better agreement if you apply these stricter criteria to your analysis?*

We do not have the information to carry out such a study. This would be a complete new project given the large number of measurements and measurement days.

*Fig. 1. Remove 1981 from the caption.*

I added "after".

*Fig. 3. Consider replacing by HYSPLIT trajectories.*

Already answered: we need one LAGRANTO example!

*Fig. 4. Include a similar plot for the water vapor data if available.*

Already answered: I added humidity information which is very important (RH = 1%).

*Fig. 7. If possible, use the same vertical scale on both figures. Idem in Fig. 9.*

This had been considered. It is not necessary for Fig. 9. For Fig. 7 I gave up this idea because an ugly white area would be created below 3 km in the $H_2O$ figure.

**Anonymous Referee #3**

*This paper begins with the surprising title, "The underestimated role of stratosphere-totroposphere transport on tropospheric ozone" which implies that all previous estimates of the global STE budget over the past 20-30 years are far too low. If supported by solid evidence, this would be a landmark paper that would force a major reassessment of the global models that quantify the tropospheric ozone budget. This is an extraordinary claim and extraordinary claims require extraordinary evidence. This paper provides no such evidence as it merely reports measurements from a single location followed by speculation, incorrect assumptions and absolutely no convincing arguments that previous STE budgets are too low. In addition, the study's major conclusions were discovered and reported by other well-known papers 18-19 years earlier using more extensive datasets. I firmly recommend that this paper be rejected.*

We are no heading for a landmark paper although I think our data are very interesting indeed. Our arguments are not based on a single location, although our high stratospheric fractions are a true highlight. I did include papers from the past 20 years, now even more. Thus, the effort is not at all only limited to a single location, but more observations for other regions should be analysed. I re-organized the Introduction for more clarity and added some information. Most importantly, it is important not to confuse STE and STT. STE means looking at both STT and TST. To evaluate the STE budget is what the models can deliver. At ground-based stations a quantification of TST is not easily possible although strong efforts to measure lower-stratospheric water vapour are made within the global NDACC network.

I have had problems with finding a less misleading title. This title was produced by quick inspiration when submitting the abstract for my Edinburgh conference. For sure I want to avoid a boring title including "at Garmisch-Partenkirchen". I now added a question mark which acknowledges that there are open questions. One important question is that about trends at sites outside Europe.

*This review is framed around the following question: Which studies, specifically, have underestimated the role of stratosphere-to-troposphere transport on tropospheric ozone? 1) The global role of STE on the tropospheric ozone budget can only be estimated by models as the current observational network is far too limited to provide global coverage. There are many well-known studies in the literature on this topic. Stevenson et al. (2006) and Young et al. (2013) provide estimates from model ensembles, with Stevenson et al. estimating the STE global ozone flux to be approximately 550 Tg, with a range of 400-800 Tg, and Young et al. giving a slightly narrower range of 400-660 Tg. Recently, three new studies (Olsen et al., 2013; Yang et al., 2016; Jaeglé et al., 2017) have received a lot of attention from the modelling community because they estimate the STE ozone flux using the latest NASA global reanalyses that are available at much higher resolution than the chemistry-climate model simulations reported by Stevenson and Young. These reanalyses also have improved dynamics with excellent coupling between the stratosphere and the troposphere, and they assimilate satellite observations to produce accurate estimates of the stratospheric ozone burden. For example a recent paper by Knowland et al. (2017) demonstrates that the NASA MERRA-2 reanalysis accurately simulates deep stratospheric intrusions above North America due to its very high horizontal resolution of 50 km and 72 vertical layers. The average STE net ozone fluxes of the three budget studies mentioned above are 489 (Olsen), 493 (Jaeglé) and 448 Tg (Yang), similar to previous estimates reported by Stevenson and Young and similar to other estimates made in the 1990s. Essentially, the STE net ozone flux has been estimated to be roughly 500 Tg for the past 20 years, regardless of the methodology. What evidence do the authors have that all of these studies have underestimated the STE net ozone flux?*

I appreciate your discussion of recent literature. I read again most of these papers. As you mention: good agreement is found concerning STE. The STE rate is only 10 % of the photochemical production rate!!! I added a few more words about the modelling results.

However, STE is not what our paper is about! In particular deep STT is, still, an issue for models as explicitly discussed. It is not only high resolution that matters, it is also the mixing concept that must be revised. I add an example of an ECMWF high-resolution water-vapour profile in comparison with lidar and sonde measurements (from (Trickl et al., 2016)):

[Figure]

There is no agreement with the measured reality at all! In addition, I have not yet seen any modelling paper reproducing the Zugspitze ozone trend.

*2) Despite the wide availability of publications that provide estimates of the STE net ozone flux, this paper only focuses on a single estimate of the ozone flux, a very old publication by Roelofs and Lelieveld (1997) that uses a relatively primitive global model that was developed in 1995, with very coarse resolution of 3.75 degrees. The authors seem to approve of this model because it estimates that 40% of the tropospheric ozone burden is from the stratosphere, which agrees with the authors' estimate of stratospheric ozone above southern Germany. But the estimate by Roelofs and Lelieveld is averaged over the entire globe and across all seasons and cannot be applied to any particular location. They state that the quantity of stratospheric ozone in the troposphere varies with altitude, latitude and season. The fact that this 40% value matches the value above southern Germany is merely coincidental and does not mean that the observations from a single location can provide meaningful evaluation of an average global estimate.*

These remarks are really unbelievable. This is all very obvious. In any case, I added a statement mentioning the high uncertainty of this result. I meanwhile learnt from D. Stevenson that STT component of tropospheric ozone is usually determined by a semi-Lagrangian approach. In the two publications specifying a STT percentage available to me this is, indeed, the case. But both papers are based on coarse-resolution modelling. However, there is no doubt that assessments based on observations

*Even if the authors could evaluate the Roelefs and Lelieveld estimate at multiple locations and across all seasons and conclude that their estimate is correct, how would this prove that previous estimates of STE are too low? Roelofs and Lelieveld's estimate of the net global flux of stratospheric ozone into the troposphere is 459 Tg per year, and their estimate of gross photochemical ozone production is 3425 Tg per year, a factor of 7.5 greater than the net flux from the stratosphere. These numbers are entirely consistent with the ozone budget estimates described above by Stevenson, Young, Olsen, Yang and Jaegle. By accepting the results of Roelofs and Lelieveld, which agree with all the other studies, the authors are essentially agreeing with the consensus view that STE contributes roughly 450-500 TG of ozone per year to the troposphere. Again: Which studies, specifically, have underestimated the role of stratosphere-to-troposphere transport on tropospheric ozone?*

Again and again, you focus on STE rather than accepting that the paper is about STT! The net STE rate is, of course, much smaller than the photochemical production rate.

*3) The authors highlight the fact that their lidar detects many dry layers above southern Germany especially in summer, and claim that this is a new result. But this is not a new finding. Nearly 20 years ago the late Reginald Newell at MIT published a well known paper in Nature in 1999 (cited 126 times according to Google Scholar) with the title: "Ubiquity of quasi-horizontal layers in the troposphere". They found, using the extensive MOZAIC database of ozone and water vapor profiles that fine layers in the atmosphere are found everywhere, and that dry layers with enhanced ozone are the most common, indicating a pervasive influence from STE. A year later V. Thouret published an even more extensive analysis based on thousands of MOZAIC profiles around the world (half above western Europe), and here are some of their important findings:*

*"At northern midlatitudes we find 4 times more layers in summer than in winter, while in tropical Asia we observe a spring maximum in the occurrence of the layers. The most abundant layer type everywhere is O3+H2O− and corresponds to the signature of stratospheric intrusions or continental pollution." "The most surprising feature in this study is the lack of any seasonal variations. The only seasonal cycle and regional difference observed concern the average of the number of layers per profile (strong maximum in summer at midlatitudes and in spring in tropical Asia)." "This new finding reveals that stratospheric intrusions are not negligible in summer at midlatitudes or in the tropics, as previously thought" Therefore, the finding by Trickl et al. regarding the high frequency of ozone-rich layers in the mid-troposphere above Europe during*

*summer is nothing new. Thouret et al. reached the same conclusion 18 years earlier using a far more extensive data set and a much more thorough and clear analysis. The excellent and ground-breaking papers by Newell et al. and Thouret et al. demonstrate that since the publication date of these papers, STE has not been underestimated above western Europe or other regions of the northern hemisphere (where MOZAIC observations were available). Their findings were not mentioned by Trickl et al.*

These papers are well known to us and we cited Newell et al. in our 2011 paper in ACP. What is different in our current paper is that we are not interested in the presence of layers (a fact also mentioned by Penkett et al. in the past) and that they are "ubiquitous", but on extremely dry layers characteristic of STT. Newell et al. and Thouret et al. use a threshold of just 5 % deviation from the surrounding background RH. Our range of interest is typically 0-6 % RH. It is worth mentioning that within the troposphere there is not much instrumentation that can resolve this. Our lidar and the RS92 sonde do have this capability.

*4) HYSPLIT is a very useful Lagrangian tool for a first look at prevailing transport patterns, but for quantitative analysis such as quantifying the amount of air or ozone transported from the stratosphere to the troposphere, it is an extremely poor choice, especially when far better models such as LAGRANTO or FLEXPART are available. The authors know how to use LAGRANTO and apply it to short-term transport analysis (<= 5 days) but for some reason, they don't use this superior tool for longer time periods. Instead they simply run HYSPLIT 15-day single back trajectories that don't give any information as to when, or if, the air parcel crossed the tropopause. They also don't give any information as to the various source regions of the air parcel. As has been shown over and over again by the many FLEXPART studies, single back trajectories are inferior to so-called retroplumes (based on 10,000-20,000 or more trajectories), and for 15-day periods single trajectories are essentially useless. Just because a 15-day single back trajectory from a dry air layer nearly circles the globe, doesn't mean that the dry filament persisted for 15 days. The HYSPLIT analysis in this paper did nothing to further my understanding of STE processes.*

As is now pointed out in the section on the methods HYSPLIT is ideal for quick identifications for our large number of routine measurements. Rather late in the course of the measurements we realized the unbelievable number of intrusion layers in our data. It was not before spring 2017 that I decided to participate in the publication effort for the 2016 Ozone Symposium. There was just time to complete the routine analysis for the early years which took me several months. I agree that LAGRANTO and FLEXPART would be superior, but there has not been any funding for such an much higher effort which would had come too late anyway. As now mentioned FLEXPART retroplume calculations were just used for selected case studies because of the huge amount of information in the FLEXPART output (Trickl et al., 2011 and 2014).

*5) On page 2 line 35 the authors speculate that global climate change is responsible for a doubling of stratospheric ozone at Zugspitze (increasing from 11.3 ppb to 23.4 ppb) for the 26-year time period from 1978 to 2004, but provide no analysis to back this up. They even state that this is a reasonable explanation. To the contrary, this is not a reasonable hypothesis at all. A brief check of the latest global temperature trend provided by NASA, NOAA or the EU's Copernicus shows that the global temperature over that time period increased by 0.4 degrees C. So do the authors believe that for this relatively small increase in temperature the net global STE flux increased by 100%? What would be the mechanism? A doubling of the overturning of the Brewer-Dobson circulation? Wouldn't such an enormous change in the global circulation be clearly evident in other components of the general circulation, such as a dramatic widening of the tropics? Surely such a major change in the general circulation over such a short time period would have been extensively described in the peer-reviewed literature and would be evident in the ECMWF ERA-Interim reanalysis wind fields, the same ones used by LAGRANTO. Such an analysis has already been conducted by Skerlak*

*et al (2014) for the period 1979-2011. They find no global increase in stratosphere-to-troposphere transport. This published finding immediately disproves the speculation of the authors in the Introduction. Furthermore, an excellent paper in Nature Geoscience by Jessica Neu (NASA JPL) shows that natural fluctuations of the stratospheric circulation due to ENSO and the Quasi-biennial Oscillation (on the order of 40%) produce tropospheric ozone changes of only 2%. Perhaps there was a shift in the regional transport patterns over Europe that increased STT over 1978-2004? Based on the results of Skerlak et al. this doesn't seem to be the case either. But given that the authors of this paper know how to use LAGRANTO, and given that the LAGRANTO analysis of Skerlak et al. must be archived somewhere, why don't they simply take the Skerlak et al. model output and explore this possibility?*

I appreciate this comment and I know the two papers mentioned (they are in our list of references!). However, I do want not go into details here. I just want to stimulate more work with this remark. May the Zürich group is interested.

*6) What is the purpose of Section 3.4 Trans-Atlantic Transport? The paper is supposed to be about the underestimated role of stratosphere-to-troposphere transport on tropospheric ozone. Yet in this section they address a completely different topic which is the transport of surface ozone from North America to Europe. This is nothing new and the peer-reviewed literature is full of studies that explore this topic, studies which are far more thorough and convincing. Here the authors use a few HYSPLIT back trajectories to speculate that ozone was transported from the US boundary layer to the free troposphere above Germany. The authors speculate that the rising trajectories could be associated with a warm conveyor belt. But they provided no analysis to confirm this transport mechanism. This isn't difficult to confirm. The authors could check archived satellite imagery. The authors could also use the RAQMS model output that is shown in Figure 14. Brad Pierce has more than 20 years of experience running model analyses for aircraft campaigns all around the world. It's easy for him to quantify the contribution of the USA or STT to ozone above Germany. Instead the authors rely on just a few single HYSPLIT back trajectories. I learned nothing new from this section, and if I want to understand how ozone is transported from the USA to Europe I will consult other studies in the literature which are for more thorough and quantitative.*

This section is not about assessing trans-Atlantic transport. Again, you list well-known facts. An assessment of trans-Atlantic transport is a demanding separate project using better modelling approaches such as FLEXPART. I have done this in the past, not all material is published. The reason for adding this chapter (which is partly rewritten for more clarity) is related to the issue that STT layers co-exist with high ozone from frequently unknown source regions, which makes the analysis of both STT and other kinds of long-range transport very difficult. Given all the long-range transport studies we have done in the past we feel that these remarks could be helpful for future more comprehensive studies.

*7) In addition to the serious problems described above, in general the manuscript is poorly written and disorganized, and tends to jump from one idea to another in a very confusing manner.*

This was also criticized by the other two reviewers. However, they were very co-operative and listed all the deficiencies I missed due to the strict submission deadline.

*8) Finally, Hans-Eckhart Scheel is listed as a co-author but he died nearly five years ago. While his measurements may have been included in this paper, he was not able to contribute to the writing of the paper and therefore he should not be listed as a co-author as he is unable to approve of the data interpretation or the conclusions.*

Dr. Scheel was the true pioneer of our STT-related work. It is not only his data that matter here: Despite being located in the Introduction Fig. 1 is in principle a result, the perhaps most important

results of our former project ATMOFAST. I finally decided to make this figure available in the reviewed literature. I am going to ask the editor for his opinion.

***References:***

*Jaeglé, L. et al (2017), Multiyear composite view of ozone enhancements and stratosphere-to-troposphere transport in dry intrusions of northern hemisphere extratropical cyclones. Journal of Geophysical Research, 122. https://doi.org/10.1002/2017JD027656*

*Knowland, K. et al. (2017). Stratospheric intrusion-influenced ozone air quality exceedances investigated in the NASA MERRA-2 reanalysis, Geophysical Research Letters, 44, 10,691–10,701. https://doi.org/10.1002/2017GL074532*

*Neu, J. et al. (2014) Tropospheric ozone variations governed by changes in stratospheric circulation, Nature Geo., 7, 340-344.*

*Newell, R.E., Thouret, V., et al. (1999). Ubiquity of quasi-horizontal layers in the troposphere. Nature, 398(6725), p.316.*

*Olsen, M. A., Douglass, A. R., & Kaplan, T. B. (2013). Variability of extratropical ozone stratosphere–troposphere exchange using microwave limb sounder observations. Journal of Geophysical Research: Atmospheres, 118, 1090–1099. https://doi.org/10.1029/2012JD018465*

*Škerlak, B., M. Sprenger, and H. Wernli [2014], A global climatology of stratospheretroposphere exchange using the ERA-Interim data set from 1979 to 2011, Atmos Chem Phys, 14, 913–937, doi:10.5194/acp-14-913-2014.*

*Stevenson, D. et al. [2006]. Multimodel ensemble simulations of present-day and near future tropospheric ozone, J. Geophys. Res. 111(D8).*

*Thouret, V., Cho, J.Y., Newell, R.E., Marenco, A. and Smit, H.G., 2000. General characteristics of tropospheric trace constituent layers observed in the MOZAIC program. Journal of Geophysical Research: Atmospheres, 105(D13), pp.17379-17392.*

*Yang, H. et al [2016], Quantifying isentropic stratosphere-troposphere exchange of ozone, J. Geophys. Res. Atmos., 121, 3372–3387; doi:10.1002/2015JD024180*

*Young, P. et al. [2013] Pre-industrial to end 21st century projections of tropospheric ozone from the Atmospheric Chemistry and Climate Model Intercomparison Project (ACCMIP), Atmos Chem Phys, 13, 2063–2090;* doi:10.5194/ acp-13-2063-2013

Thank you for these references! All but two are known to me. You may be aware that many of these publications are cited in our paper, now also the MOZAIC papers. I added the first one (Jaegle et al.) since this paper derives numbers for STT.